JCB Journal of Cell Biology

# An integrin axis induces IFN-β production in plasmacytoid dendritic cells

Davina Camargo Madeira Simoes[1,2]*, Nikolaos Paschalidis[1]*, Evangelia Kourepini[1]**, and Vily Panoutsakopoulou[1]**

Type I interferon (IFN) production by plasmacytoid dendritic cells (pDCs) has been mainly studied in the context of Toll-like receptor (TLR) activation. In the current report, we reveal that, in the absence of TLR activation, the integrin-binding SLAYGLR motif of secreted osteopontin (sOpn) induces IFN-β production in murine pDCs. This process is mediated by α4β1 integrin, indicating that integrin triggering may act as a subtle danger signal leading to IFN-β induction. The SLAYGLR-mediated α4 integrin/IFN-β axis is MyD88 independent and operates via a PI3K/mTOR/IRF3 pathway. Consequently, SLAYGLR-treated pDCs produce increased levels of type I IFNs following TLR stimulation. Intratumoral administration of SLAYGLR induces accumulation of IFN-β–expressing pDCs and efficiently suppresses melanoma tumor growth. In this process, pDCs are crucial. Finally, SLAYGLR enhances pDC development from bone marrow progenitors. These findings open new questions on the roles of sOpn and integrin α4 during homeostasis and inflammation. The newly identified integrin/IFN-β axis may be implicated in a wide array of immune responses.

## Introduction

Plasmacytoid dendritic cells (pDCs) comprise a subset that specializes in the production of type I IFNs following virus or endogenous nucleic acid recognition through Toll-like receptor 7 (TLR7) and TLR9 (Blasius and Beutler, 2010; Gilliet et al., 2008; Kawai and Akira, 2011). pDCs promote antiviral immune responses and have been implicated in the pathogenesis of autoimmune diseases, such as systemic lupus erythematosus, that are characterized by a type I IFN cytokine signature (Ganguly et al., 2013). In addition, pDCs can also protect from melanoma, and this effect is mediated by type I IFNs (Drobits et al., 2012).

Type I IFN production by pDCs has been mainly studied in the context of activation of TLRs by pathogen-associated molecular patterns (PAMPs; Blasius and Beutler, 2010; Gilliet et al., 2008; Kawai and Akira, 2011). An elegant study has shown that recognition of type A cytosine-phosphorothioate-guanine (CpG-A) oligodeoxynucleotides (ODNs) by TLR9 in pDCs results in robust production of IFN-α by a process dependent on the crucial interaction of the adaptor molecule myeloid differentiation primary response protein 88 (MyD88) with the intracellular form of osteopontin (iOpn; Shinohara et al., 2006). A secreted form of osteopontin (sOpn) is also expressed by immune and nonimmune cells (Chabas et al., 2001; Wang and Denhardt, 2008). Previous work has shown that

during TLR9 triggering, sOpn had no effects on IFN-α production by pDCs, while its effects on IFN-β levels were not tested (Shinohara et al., 2006). In addition, as sOpn is constitutively expressed (Grassinger et al., 2009), its effects on pDCs and their type I IFN production, in the absence of known danger signals, remain elusive.

In inflammatory conditions, sOpn affects DC function, influencing various adaptive immune responses (Kawamura et al., 2005; Kourepini et al., 2014; Murugaiyan et al., 2010; Renkl et al., 2005; Shinohara et al., 2008; Shinohara et al., 2006; Xanthou et al., 2007), as well as pDC recruitment to the draining lymph nodes (Xanthou et al., 2007). Under noninflammatory conditions and homeostasis, sOpn is constitutively expressed by a great variety of cells (Chiodoni et al., 2010; Gerstenfeld, 1999; Uede, 2011; Wang and Denhardt, 2008); however, its physiological significance is largely unknown. In the bone marrow (BM), sOpn is expressed mainly in the form of thrombin-cleaved fragments (Grassinger et al., 2009). Opn fragment containing the binding-motif SLAYGLR, which is revealed after thrombin cleavage, interacts with α4β1-, α9β1-, and α4β7-integrins, whereas the RGD motif interacts with αvβ3-, αvβ5-, αvβ1-, and α5β1-integrins (Chabas et al., 2001; Wang and Denhardt, 2008).

In the current report, we reveal that, in the absence of PAMPs or antigens, sOpn—and specifically its integrin-binding

[1]Cellular Immunology Laboratory of Vily Panoutsakopoulou, Center for Basic Research, Biomedical Research Foundation of the Academy of Athens, Athens, Greece; [2]Faculty of Health and Life Sciences, Northumbria University Newcastle, Newcastle upon Tyne, UK.

Dr. Panoutsakopoulou died on November 8, 2018. *D.C.M. Simoes and N. Paschalidis contributed equally to this paper; **V. Panoutsakopoulou and E. Kourepini contributed equally to this paper. Correspondence to Evangelia Kourepini: ekourepini@bioacademy.gr; Davina Camargo Madeira Simoes: davina.simoes@northumbria.ac.uk.

SLAYGLR motif—induces low levels of IFN-β expression in pDCs. This process is mediated by α4β1-integrin triggering, indicating that integrins alone can act as subtle danger signals leading to IFN-β induction in pDCs. We find that this sOpn/IFN-β axis is MyD88 independent but operates via phosphoinositide 3-kinase (PI3K)/mammalian target for rapamycin (mTOR)/IFN regulatory factor (IRF)3 activation. In addition, Opn/α4 integrin preactivated pDCs are predisposed to produce increased levels of type I IFN following TLR stimulation. Upon development of melanoma, intratumoral administration of Opn/SLAYGLR recruits in situ efficient pDCs competent in suppressing tumor growth. Finally, integrin-binding SLAYGLR exerts a fostering effect on Flt3L-mediated pDC development from BM progenitors, a process that could also lead to the increment of tumor-suppressive immune response.

## Results

### The integrin-binding SLAYGLR motif of Opn induces IFN-β production by pDCs under pathogenic/antigenic-free conditions

When we treated isolated pDCs (7AAD$^-$CD3$^-$CD19$^-$CD11c$^+$CD11b$^-$B220$^+$PDCA-1$^+$Siglec-H$^+$) with endotoxin-free recombinant Opn (rOpn), IFN-β mRNA and protein levels were significantly increased compared with control-treated pDCs. By using Opn fragment peptides (frOpn), we determined that this increment of IFN-β expression in the rOpn-treated pDCs was attributed to frOpn1, which contains an intact SLAYGLR domain (Fig. 1 a). IFN-β elevation was consistent with our previous study, in which frOpn1 induced IFN-β production from endotoxin-free antigen-primed pDCs (Alissafi et al., 2018). In fact, frOpn1 was even more potent than rOpn in inducing IFN-β production by pDCs, whereas frOpn2 (intact RGD) and frOpn3 (scrambled) were unable to increase IFN-β production (Fig. 1 a). Furthermore, by utilizing a very sensitive ELISA kit, we found that frOpn1 could elicit expression of IFN-β protein by pDCs, while there was no induction of IFN-α (Fig. 1 b). We verified the enhancing effect of frOpn1 on Ifnb expression by using isolated pDCs from Ifnb$^{EYFP}$ reporter mice (Fig. S1 a). IFN-β induction by frOpn1 was independent of the type I IFN receptor IFNAR1, as it was successfully induced in Ifnar1$^{-/-}$ pDCs (Fig. 1 b). Again, IFN-α production was not detected in Ifnar1$^{-/-}$ pDCs treated with frOpn1 (Fig. 1 b). Finally, higher concentrations of frOpn1 could elicit a dose response in terms of IFN-β production by pDCs (Fig. 1 c). Also, during BM-Flt3L pDC generation, addition of either rOpn or frOpn1 induced increased expression of Ifnb in isolated pDCs compared with PBS (Fig. 2 b). As the type I IFN-Flt3L axis is important for the generation of pDCs from BM progenitor cells (Chen et al., 2013), we evaluated whether Opn had an effect on in vitro development of pDCs and conventional DCs (cDCs). Treatment of BM Flt3L cell cultures with either rOpn or frOpn1 resulted in a significant increment in the numbers of pDCs and a concomitant reduction in the numbers of cDCs compared with PBS treatment (Fig. 2 a). In Flt3L-deprived BM cell cultures, similarly rOpn or frOpn1 resulted in a significant increase in pDC number and decrease in cDC number compared with PBS (Fig. 2 a). Thus, Opn in Flt3L

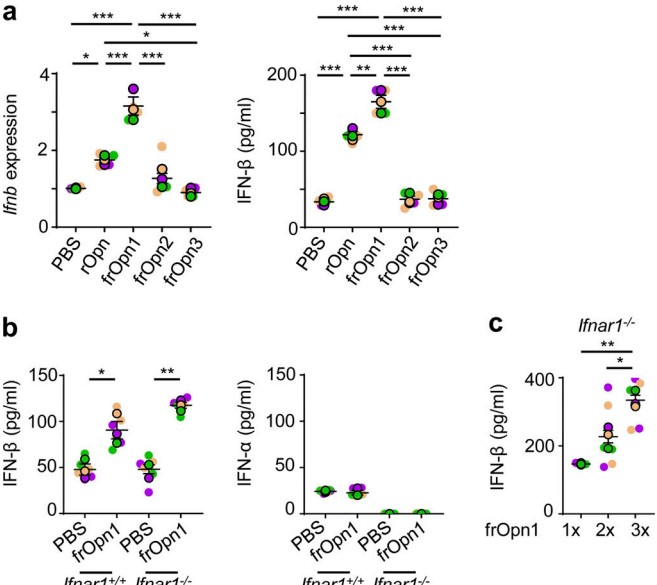

Figure 1. **The SLAYGLR motif of Opn upregulates IFN-β production by pDCs. (a)** Relative Ifnb mRNA expression of BM-derived pDCs isolated and cultured with either rOpn or Opn fragments: frOpn1, or frOpn2, or frOpn3 or PBS (control), and IFN-β secretion levels in culture supernatants of the same cell cultures. Data are presented as mean ± SEM (n = 4 wells per group) from three independent experiments. **(b)** IFN-β and IFN-α levels in the supernatants of Ifnar1$^{+/+}$ and Ifnar1$^{-/-}$ BM-derived pDCs cultured with either frOpn1 or PBS control. Data are presented as mean ± SEM (n = 6 wells per group) from three independent experiments. **(c)** Levels of IFN-β production by Ifnar1$^{-/-}$ pDCs stimulated with increasing amounts of frOpn1. Data are presented as mean ± SEM (n = 3–6 wells per group) from three independent experiments. Each experimental replicate is presented with different-colored dots, and dots with black line borders are the averages derived from each replicate. *, P ≤ 0.033; **, P ≤ 0.002; ***, P ≤ 0.001 (two-way ANOVA with Bonferroni's multiple comparison test).

cultures switches the DC balance toward pDCs. All the above indicate that the integrin-binding SLAYGLR motif of Opn induces IFN-β expression in pDCs in the absence of danger signals.

### Opn/SLAYGLR activates IRF3 via integrin α4

Activation of IRF3 and IRF7 is fundamental for type I IFN production (Sato et al., 2000). We initially observed that frOpn1 treatment of Ifnar1$^{-/-}$ pDCs induced significant increase in Irf3 expression after 3 h of treatment, while Irf7 expression remained the same (Fig. 3 a). At the same time point, frOpn1 also boosted phosphorylated and total IRF3 protein levels compared with control PBS-treated group (Fig. 3 b), whereas IRF7 phosphorylation was below the detectable range in our settings. Confocal microscopy analysis revealed that total cellular IRF3 and its nuclear translocation was significantly increased in frOpn1-treated Ifnar1$^{-/-}$ pDCs, compared with the control group (Fig. 3 c). As the SLAYGLR motif of Opn is known to interact with α4β1 and α9β1 integrins (Wang and Denhardt, 2008) expressed by pDCs, we asked which of the two integrins was involved in frOpn1-mediated IRF3 nuclear translocation. Blockade of α4 integrin with a blocking antibody significantly reduced IRF3 nuclear translocation in frOpn1-treated pDCs, whereas

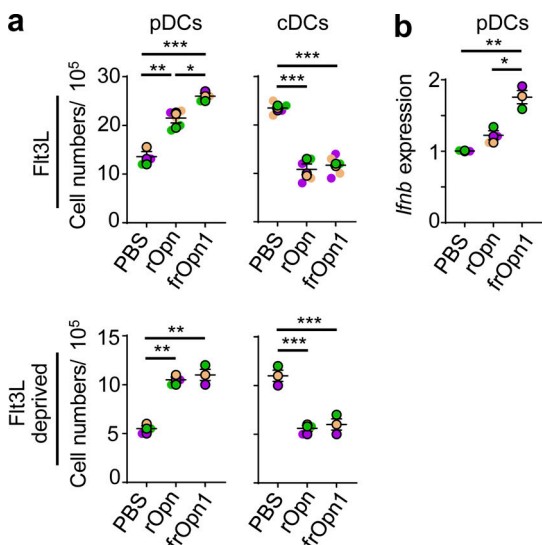

**a**

pDCs    cDCs

**b**

pDCs

Figure 2. **Opn affects pDC and cDC generation.** BM cells were isolated and cultured with Flt3L or in Flt3L-deprived conditions in the presence of rOpn, frOpn1, or PBS (control). **(a)** Numbers of 7AAD⁻CD11c⁺CD11b⁻PDCA-1⁺Siglec-H⁺ pDCs and 7AAD⁻CD11c⁺CD11b⁺PDCA-1⁻ cDCs generated from BM cells with or without Flt3L and rOpn, frOpn1, and PBS. **(b)** Relative *Ifnb* gene expression in BM-derived pDCs. Data are presented as mean ± SEM (*n* = 3–5 wells per group) from three independent experiments (a and b). Each experimental replicate is presented with different-colored dots, and dots with black line borders are the averages derived from each replicate. *, P ≤ 0.033; **, P ≤ 0.002; ***, P ≤ 0.001 (two-way ANOVA with Bonferroni's multiple comparison test).

blockade of α9 integrin did not compromise IRF3 nuclear translocation (Fig. 3 d). Blockade of α4 integrin resulted in a twofold reduction in *Ifnb* expression of frOpn1-treated pDCs, whereas upon α9 integrin blockade, *Ifnb* expression was not decreased (Fig. 3 e). Furthermore, blocking of α4β1 integrin significantly reduced secretion of IFN-β by pDCs (Fig. 3 f). Therefore, the SLAYGLR motif of Opn acts on α4 integrin, triggering IFN-β production.

To survey whether IRF3 is directly necessary for the induction of IFN-β in response to Opn, we knocked down IRF3 in pDCs by siRNA. We verified silenced IRF3 gene expression in pDCs by RT-PCR (Fig. S1 b), diminished IRF3 protein by immunoblotting (Fig. S1 c), and validated loss-of-function by measuring IFN-β production upon poly(I:C) stimulation of pDCs (Fig. S1 d). In fact, IRF3-silenced pDCs exhibited diminished ability to enhance both expression and production of IFN-β after frOpn1 treatment (Fig. 3 g). Thus, IRF3 is a crucial mediator of the Opn-induced IFN-β production observed via integrin α4.

### Opn/SLAYGLR activation of IRF3 and IFN-β production do not depend on MyD88

During TLR9 activation of pDCs, iOpn has been shown to enhance IFN-α production in a MyD88-dependent manner (Shinohara et al., 2006). Although we do not use TLR triggers in our system, we examined the role of MyD88 for frOpn1-induced production of IFN-β. Notably, endotoxin-free frOpn1 treatment of *Myd88⁻/⁻* pDCs induced a significant increase in secretion of

IFN-β compared with controls and *Myd88⁺/⁺* pDCs (Fig. 4 a). This enhancement in IFN-β secretion was not the result of an autocrine loop (Barchet et al., 2002), as levels of IFN-β were also increased in *Ifnar1⁻/⁻* pDCs treated with MyD88 inhibitor and frOpn1 or PBS, compared with control vehicle– and control peptide–treated pDCs (Fig. 4 b). In fact, it appears that frOpn1 is a more potent inducer of IFN-β when MyD88 is blocked or absent (Fig. 4, a and b). As frOpn1 triggered a MyD88-independent increment in IFN-β secretion, we checked the IRF3 nuclear translocation upon MyD88 blockade. MyD88 inhibition induced significantly increased IRF3 nuclear translocation in frOpn1-treated *Ifnar1⁻/⁻* pDCs (Fig. 4 c). Moreover, TIR domain–containing adapter-inducing IFN β (TRIF) inhibition had no effects on *Ifnb* expression and secretion by frOpn1-treated *Ifnar1⁻/⁻* pDCs (Fig. 4 d). Therefore, frOpn1 boosts IFN-β production via a molecular pathway independent of the adaptor molecules MyD88 and TRIF.

### Opn/SLAYGLR induces IFN-β production in pDCs via the PI3K/mTOR/IRF3 axis

Opn ligation to integrins is known to activate a cascade of kinases (Ogata et al., 2007). pDC treatment with frOpn1 resulted in phosphorylation of protein kinase B (Akt), which was significantly increased at 30 min compared with 10 min after treatment and compared with control PBS treatment (Fig. 5 a). PI3K inhibition by wortmannin reduced IFN-β expression levels in frOpn1-treated *Ifnar1⁻/⁻* pDCs, as well as in *Akt1⁻/⁻* pDC cultures (Fig. 5 b). Activated Akt is known to destabilize the tuberous sclerosis complex (TSC1 and TSC2) leading to mTOR activation (Testa and Tsichlis, 2005). Thus, frOpn1-treated pDCs exhibited enhanced phosphorylation of Tsc2, significantly increased at 30 min compared with 10 min after treatment and compared with control PBS treatment (Fig. 5 a). In pDCs, mTOR can also be activated by factors that limit *Tsc1* mRNA abundance (e.g., miR-126), leading to enhanced type I IFN production (Agudo et al., 2014; Weichhart et al., 2015). In our system, frOpn1 treatment resulted only in reduction of *Tsc1* expression in pDCs (Fig. 5 c), and not in *Tsc2* expression (Fig. S2). The expression of *Tsc1* appeared to be regulated by the PI3K signaling cascade, as inhibition with wortmannin induced a fourfold increase in *Tsc1* mRNA (Fig. 5 c). Also, *Tsc1* was increased in *Akt1⁻/⁻* and *Akt2⁻/⁻* pDCs (Fig. 5 c).

As TSC1 and 2 negatively regulate mTOR/p70S6 (Weichhart et al., 2015), we found that phosphorylation of p70S6 kinase, an mTOR downstream target, was significantly elevated at both Ser371 and Thr389 1, 2, 4, and 6 h after frOpn1 treatment compared with the respective time points in the control PBS group (Fig. 5 d). The most significant rise in phosphorylation of both Ser371 and Thr389 was 2–4 h after frOpn1 treatment (Fig. 5 d). In addition, the mTOR inhibitor, rapamycin, caused a significant inhibition of IFN-β production in frOpn1-treated pDC cultures (Fig. 5 e). Rapamycin significantly reduced the expression of IRF3 and its nuclear translocation (Fig. 5, f and g). Wortmannin and Akt inhibitor were also effective at suppressing IRF3 expression and nuclear translocation in frOpn1-treated *Ifnar1⁻/⁻* pDCs (Fig. 5, f and g). In summary, the above results provide evidence that the PI3K/Akt/mTOR pathway is involved in IRF3

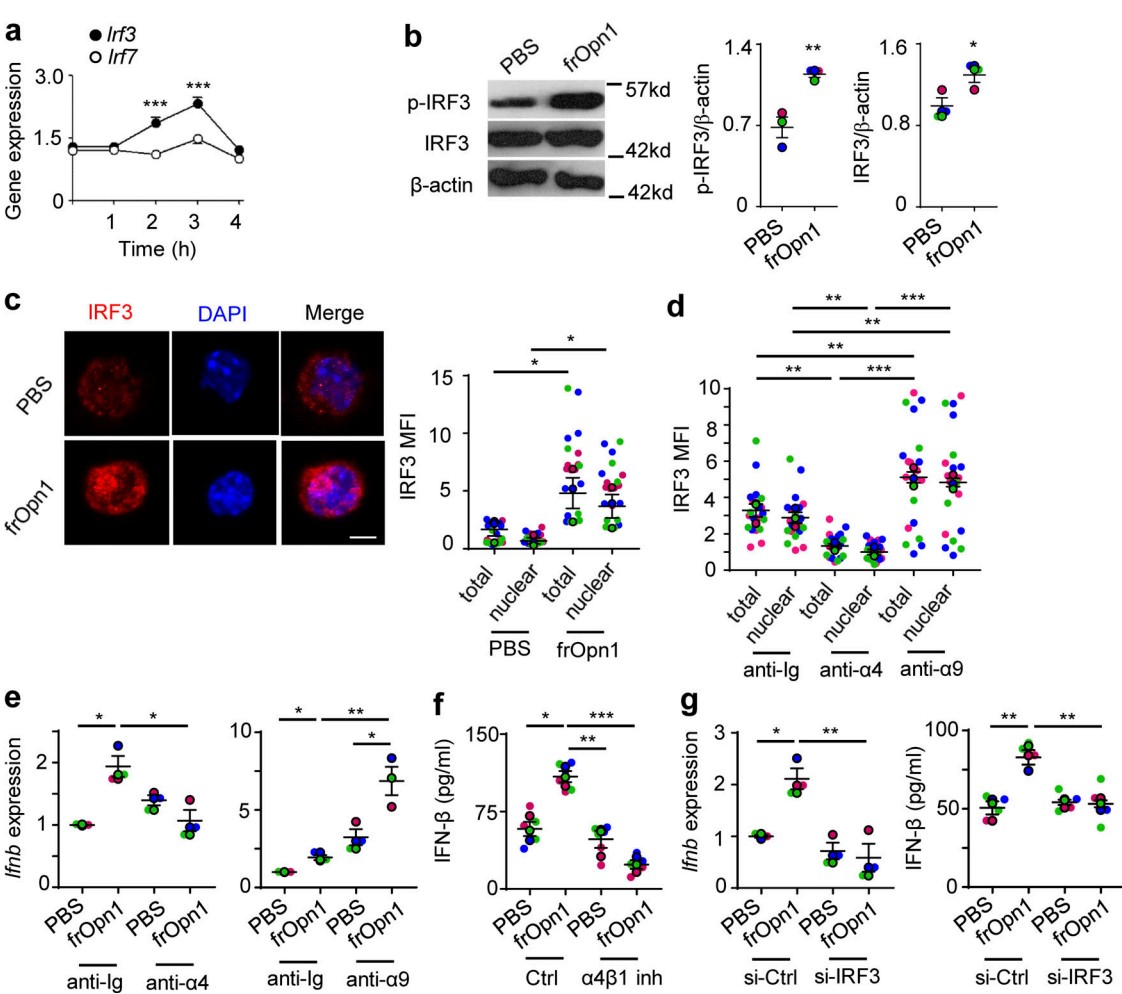

Figure 3. **Opn/SLAYGLR induces α4-integrin–mediated activation of IRF3/IFN-β.** $Ifnar1^{-/-}$ pDCs were treated in vitro with frOpn1 or PBS (control). **(a)** Relative Irf3 and Irf7 mRNA expression at different time points after frOpn1 treatment. Data are presented as mean ± SEM ($n$ = 6 wells per group) from three independent experiments. ***, P ≤ 0.001 (two-way ANOVA with Bonferroni's multiple comparison test). **(b)** Representative immunoblot of p-IRF3 and total IRF3 in pDCs after 3 h and respective quantification dot plots. Quantification was normalized to β-actin. Data are presented as mean ± SEM ($n$ = 3 wells per group) from three independent experiments. *, P ≤ 0.033; **, P ≤ 0.002 (unpaired two-tailed Student's $t$ test). **(c and d)** Immunofluorescence confocal microscopy analysis of IRF3 (Alexa Fluor 647, red) and DAPI (blue) quantified in ≥20 pDCs treated for 3 h. **(c)** Representative images of treated pDCs. Scale bar, 5 μm. Dot plot graph depicts corresponding individual cell total (nuclear + cytoplasmic) and nuclear IRF3 MFI. Each dot (without border) represents one cell measurement. Data are presented as mean ± SEM ($n$ = 21 cells per group) from three independent experiments. **(d)** MFI of total and nuclear IRF3 when α4/α9 blocking or Ig control antibodies were added before the 3-h frOpn1 treatment. Each dot (without border) represents one cell measurement. Data are presented as mean ± SEM ($n$ = 20 cells per group) from three independent experiments. **(e)** Relative $Ifnb$ mRNA expression in pDCs upon α4/α9 blocking or Ig control antibody addition before a 3-h frOpn1 treatment or PBS. **(f)** Levels of IFN-β secretion in the supernatants of pDC cultures added with α4β1 inhibitor or vehicle (control) before frOpn1 or PBS treatment. **(g)** Relative $Ifnb$ mRNA expression and levels of IFN-β secretion after IRF3 knocked down in pDCs (si-IRF3), compared with control (si-Ctrl). Values were normalized relative to $Hprt$ expression (a, e, and g). Data are presented as mean ± SEM ($n$ = 3–6 per group) from three independent experiments (e–g). Each experimental replicate is presented with different-colored dots, and dots with black line borders are the averages derived from each replicate (b–g). *, P ≤ 0.033; **, P ≤ 0.002; ***, P ≤ 0.001 (two-way ANOVA with Bonferroni's multiple comparison test).

expression (Fig. 5 f) and its nuclear translocation in pDCs (Fig. 5 g). Therefore, the SLAYGLR motif of Opn activates the α4 integrin/PI3K/mTOR/p70S6K cascade leading to IRF3 activation and $Ifnb$ expression (Fig. 6 b).

We further evaluated the levels of IRF3 mRNA, phospho-IRF3, and total IRF3 over a detailed time course to answer whether IRF3 is activated directly at a posttranslational level or whether increased IRF3 activity is driven by upregulated transcription in response to Opn. We used immunofluorescence staining to monitor IRF3 translation and nuclear translocation upon frOpn1 stimulation of $Ifnar1^{-/-}$ pDCs. Cytoplasmic IRF3

expression starts rising from the baseline at 1.5 h, peaking at 2.5 h, and dropping at 3 h, which coincides with the 3-h peak of nuclear translocation (Fig. 6 a). Nuclear translocation of IRF3 starts rising at 2.5 h, peaking at 3 h (time point also demonstrated in Fig. 3, b and c; Fig. 4 c; and Fig. 5 g), and at 3.5 h almost returns to baseline expression to remain at these levels (Fig. 6 a). Similarly, $Irf3$ mRNA transcription starts rising at 1.5 h of treatment and reaches the highest levels at 3 h (also demonstrated in Figs. 3 a and 5 f); afterward, it decreases and remains unchanged for 9 h (Fig. 6 a). The timeframe for mTOR activation starts as early as 1–1.5 h, as both Akt and Tsc2 phosphorylation

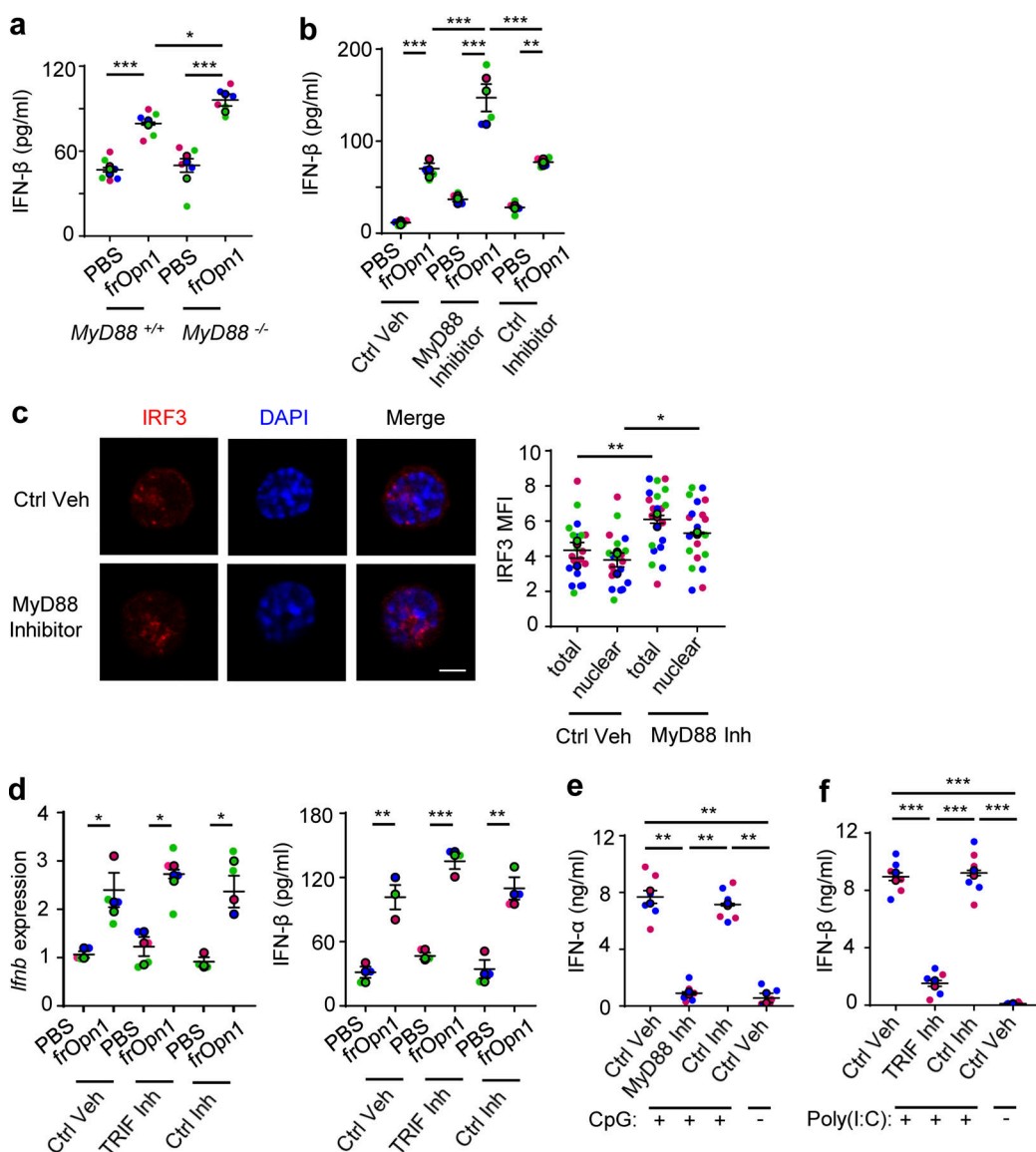

Figure 4. **Induction of IFN-β expression by Opn/SLAYGLR is MyD88 and TRIF independent.** pDCs isolated from *Ifnar1⁻/⁻* mice (unless otherwise stated) were treated in vitro with either frOpn1 or PBS (control). **(a)** Levels of IFN-β measured in supernatants of MyD88⁻/⁻- or MyD88⁺/⁺-treated pDCs. Data are presented as mean ± SEM (*n* = 6 wells per group) from three independent experiments. **(b)** Levels of IFN-β measured in supernatants of treated pDCs in the presence of Pepinh-MYD (MyD88 Inhibitor) or Pepinh-control (Ctrl Inhibitor) or vehicle (Ctrl Veh). Data are presented as mean ± SEM (*n* = 5 wells per group) from three independent experiments. **(c)** Representative images of treated pDCs added with MyD88 inhibitor or vehicle. Scale bar, 5 μm. Dot plot graph depicts total (nuclear + cytoplasmic) and nuclear IRF3 MFI in 20 representative cells. Data are presented as mean ± SEM (*n* = 20 cells per group) from three independent experiments. **(d)** Relative *Ifnb* mRNA expression in treated pDCs added with Pepinh-TRIF (TRIF Inh) or Pepinh-control (Ctrl Inh) or vehicle (Ctrl Veh), and levels of IFN-β measured in supernatants of the same cultures. Data are presented as mean ± SEM (*n* = 3–4 wells per group) from three independent experiments. **(e)** Levels of IFN-α in the supernatants of cultured pDCs after a 20-min pulse with CpG-A TLR9 agonist in the presence of Pepinh-MYD (MyD88 Inh) or Pepinh-control (Ctrl Inh) or vehicle (Ctrl Veh). **(f)** Levels of IFN-β in the supernatants of cultured pDCs treated with poly(I:C) TLR3 agonist in the presence of Pepinh-TRIF (TRIF Inh) or Pepinh-control (Ctrl Inh) or vehicle (Ctrl Veh). Data are presented as mean ± SEM (*n* = 6 wells per group) from two independent experiments (e–g). Each experimental replicate is presented with different-colored dots, and dots with black line borders are the averages derived from each replicate. *, P ≤ 0.033; **, P ≤ 0.002; ***, P ≤ 0.001 (two-way ANOVA with Bonferroni's multiple comparison test).

took place within 30 min (Fig. 5 a), and p70S6K phosphorylation started to exhibit statistically significant differences within 1 h after frOpn addition (Fig. 5 d). Therefore, IRF3 phosphorylation is not possibly a result of a direct activating effect of mTOR, as the nuclear translocation of IRF3 takes place after 2.5 h (Fig. 6 a). The above data suggest that the activation of PI3K/Akt/mTOR increases baseline values of IRF3 by upregulated transcription, and it is not activated by mTOR directly at a posttranslational level.

## Opn/SLAYGLR-induced IFN-β predisposes pDCs for exaggerated responses to TLR triggering

To examine the role of frOpn1-induced IFN-β production by pDCs, we assessed the ability of frOpn1-pretreated (primed)

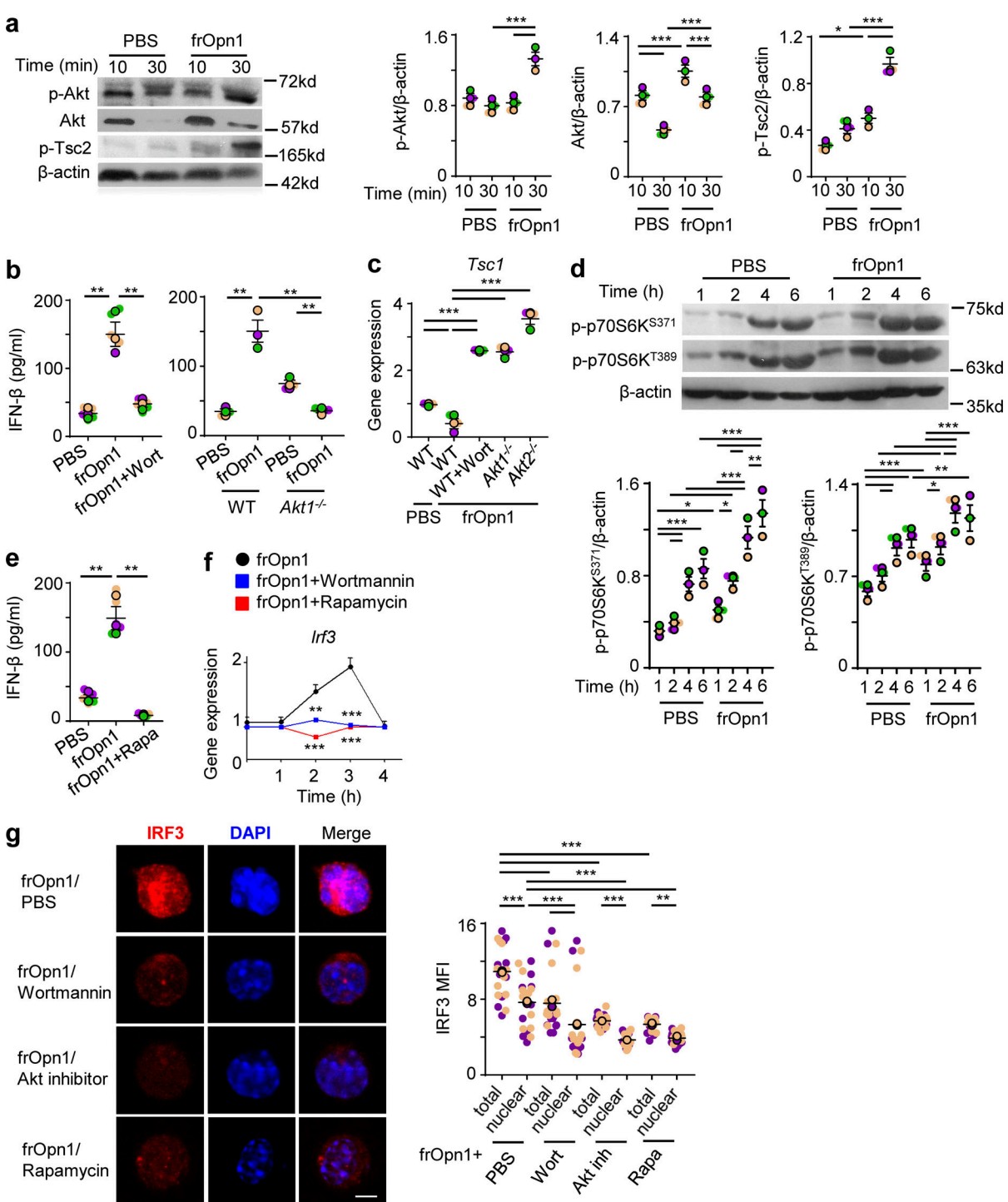

Figure 5. **Opn/SLAYGLR induces IFN-β production in pDCs via the PI3K/mTOR/IRF3 pathway.** pDCs isolated from *Ifnar1*[−/−] (unless otherwise stated) were treated in vitro with either frOpn1 or PBS (control). **(a)** Representative immunoblots of p-Akt, total Akt, and p-Tsc2 in pDCs treated for different times and respective quantification dot plots. Data are presented as mean ± SEM ($n$ = 3 wells per group) from three independent experiments. **(b)** Levels of IFN-β in the supernatants of WT pDC cultures treated with wortmannin and of *Akt1*[−/−] pDCs. Data are presented as mean ± SEM ($n$ = 3–5 wells per group) from three independent experiments. **(c)** Relative *Tsc1* mRNA expression in WT, *Akt1*[−/−], and *Akt2*[−/−] pDCs. Data are presented as mean ± SEM ($n$ = 3 wells per group) from three independent experiments. **(d)** Representative immunoblots of p-p70S6K[S371] and p-p70S6K[T389] at indicated time points and respective quantification dot plots. Data are presented as mean ± SEM ($n$ = 3 wells per group) from three independent experiments. **(e)** Levels of IFN-β in the supernatants of WT pDC cultures with added rapamycin. Data are presented as mean ± SEM ($n$ = 5 wells per group) from three independent experiments. **(f)** Relative mRNA expression of IRF3 in frOpn1-treated pDCs with added wortmannin or rapamycin at indicated time points. Data are presented as mean ± SEM ($n$ = 3 wells per group) from three independent experiments. **(g)** Representative images of frOpn1-treated pDCs after wortmannin, Akt inhibitor, or rapamycin addition. Scale bar, 5 µm. Dot plot graph depicts MFI of nuclear IRF3 immunofluorescence measured in 20 pDCs. Data are mean ± SEM ($n$ = 20 cells per group) from two independent experiments. Protein level values were normalized to β-actin (a and d). Relative mRNA values were normalized to *Hprt* expression (c and f). Each experimental replicate is presented with different-colored dots, and dots with black line borders are the averages derived from each replicate (a–e and g). *, P ≤ 0.033; **, P ≤ 0.002; ***, P ≤ 0.001 (two-way ANOVA with Bonferroni's multiple comparison test).

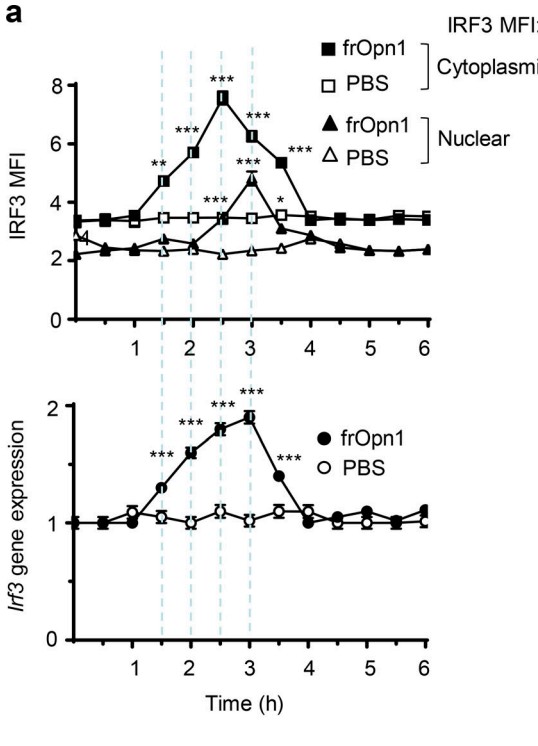

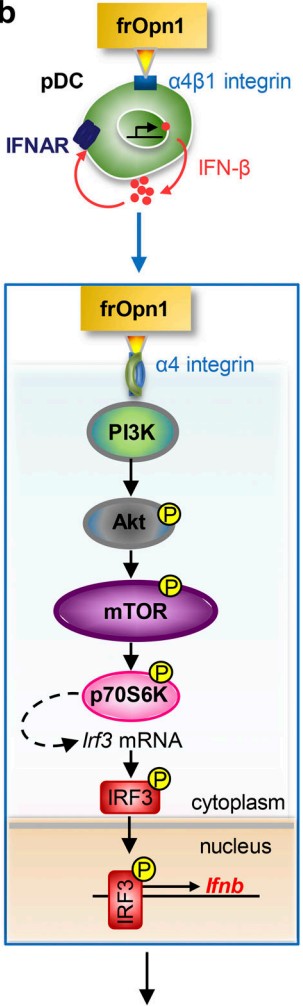

Figure 6. **Activated PI3K/Akt/mTOR increases IRF3 activity by upregulated transcription in response to Opn.** Time course of IRF3 protein and mRNA expression in Ifnar1−/− pDCs ≤6 h after frOpn1 or PBS addition (control). **(a)** Graphs depicting cytoplasmic and nuclear IRF3 MFI in 20 pDCs (top) and relative mRNA expression of *Irf3* at indicated time points, normalized to *Hprt* expression (bottom). Data are presented as mean ± SEM (n = 3 wells per group) from three independent experiments containing six mice per group. *, P ≤ 0.033; **, P ≤ 0.002; ***, P ≤ 0.001 (two-way ANOVA with Bonferroni's multiple comparison test). **(b)** Graphical abstract. Ligation of SLAYGLR domain of Opn (frOpn1) to α4 integrin enhances IFN-β production: Integrin triggers PI3K activation leading to Akt phosphorylation that activates mTOR. Posttranslational activation of mTOR signaling pathway and phosphorylation of p70S6 kinase lead to enhanced *Irf3* mRNA transcription. IRF3 protein is phosphorylated and translocated to the nucleus, where it enhances *Ifnb* gene transcription. Ligation of frOpn1 enhances IFN-β+ intratumoral pDC and IFN-γ+CD8+ T cell numbers, consequently inducing reduction of melanoma tumor size. Undetermined signaling pathway is represented by a dotted line.

pDCs to respond to TLR9 ligands. pDCs primed with frOpn1 and subsequently activated by CpG-A produced significantly increased levels of IFN-α compared with control priming (Fig. 7). IFN-α production was dependent on a positive feedback loop, as CpG-A–activated *Ifnar1*−/− pDCs did not produce detectable levels of IFN-α (Fig. 7). Also, IFN-α production was dependent on IF-NAR signaling triggered by IFN-β, as CpG-A–activated anti-IFN-β–treated pDCs did not produce detectable levels of IFN-α, compared with control anti-Ig–treated pDCs (Fig. 7). Therefore, Opn/SLAYGLR induces an initial increment in the production of IFN-β (Figs. 1, 2, 3, 4, and 5) that primes pDCs to elicit fast and robust expression of type I IFNs in response to CpG.

### The SLAYGLR fragment of Opn induces a pDC-mediated antitumor immune response

To examine the possible effects of frOpn1 on the antitumor responses of pDCs, we used B16-F10 melanoma cells, which induce aggressively growing tumors when injected subcutaneously in mice. Daily injection with frOpn1 in tumors of melanoma-bearing mice resulted in significantly enhanced recruitment of pDCs and especially IFN-β–producing pDCs, compared with scrambled frOpn3, PBS control, and pDC-depleted groups (Fig. 8,

a and b). Also, pDCs from the frOpn1-treated mice expressed more IFN-β per cell than pDCs from frOpn3- and PBS-treated mice (Fig. 8 c). To track the time point of pDC recruitment after the onset of frOpn1 administration, kinetic analysis was performed revealing the highest IFN-βEYFP+ pDC numbers in tumors on day 4, but no significant increase with frOpn3 injection (Fig. 8 d). Significant differences in IFN-βEYFP+ pDC numbers between frOpn1 and frOpn3, PBS control, and diphtheria toxin (DT) groups were observed as early as the first day after injection, and also on days 3–6 (Fig. 8 d). The addition of DT control ≤6 d after the first frOpn1 injection demonstrated the efficacy of intratumoral pDC depletion in DT-treated mice (Fig. 8 d). Apart from the pDCs, which were depleted upon DT administration (Fig. 8 d), we measured no significant changes in numbers of other immune cell subsets (Fig. S3 b). Kinetic analysis revealed that the changes in pDC numbers and their massive recruitment on day 4 reflected significantly reduced tumor growth on days 4 and 5 with frOpn1, compared with frOpn3 administration and pDC-depleted groups (Fig. 8 e). Differences were similar when measuring the actual tumor volume in all groups ≤10 d after the first frOpn1 injection (Fig. 8 e). This effect was pDC mediated, since there was no reduction in tumor growth in *BDCA2-DTR*

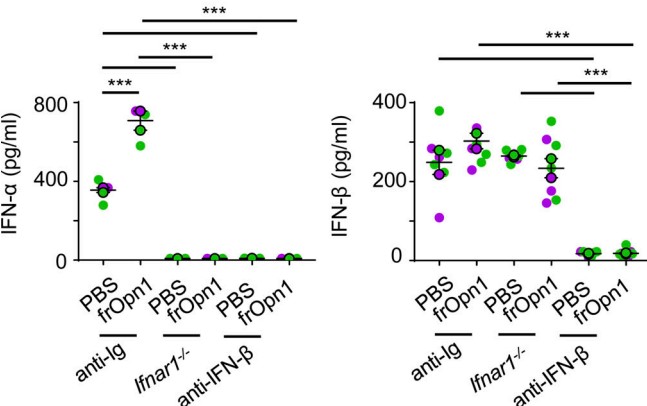

pDCs primed with: PBS or frOpn1 followed by +CpG-A treatment

Figure 7. **Opn/SLAYGLR-induced IFN-β in pDCs predisposes to exaggerated responses to TLR triggering.** Isolated *Ifnar1^+/+^* and *Ifnar^-/-^* pDCs were treated in vitro with frOpn1 or PBS (control). After a washing step, pDCs were activated with 1 µg/ml CpG-A. *Ifnar1^+/+^* pDC cultures were also added with either anti–IFN-β or anti-Ig control. IFN-α/β was measured in supernatant cultures after 24 h of incubation. Data are presented as mean ± SEM (n = 3–7 wells per group) from two independent experiments. Each experimental replicate is presented with different-colored dots, and dots with black line borders are the averages derived from each replicate. ***, P ≤ 0.001 (two-way ANOVA with Bonferroni's multiple comparison test).

transgenic mice treated with frOpn1 upon in vivo pDC depletion (Fig. 8 e). We noticed also that pDC depletion does not affect tumor growth independently of frOpn1 (Fig. 8 e). Moreover, frOpn1 administration resulted in enhanced CD8^+ T cell percentages into the tumor on day 4 after the onset of injections and also greater numbers of IFN-γ producers within the CD8^+ T cell population compared with frOpn3, frOpn1 + DT, and frOpn3 + DT administration (Fig. 8 f). The recruitment/cellularity of other tumor-infiltrating leukocyte subsets remained unchanged on day 4 after frOpn1 injection (Fig. S3 a). There was no enhancement on IFN-γ–expressing intratumoral CD8^+ T cell numbers in pDC-depleted groups, and the measurements from frOpn1 + DT and frOpn3 + DT administration were similar, implying that this increase after frOpn1 treatment was pDC mediated (Fig. 8 f). Comparison between groups with and without DT treatment revealed that pDC depletion did not affect IFN-γ–expressing intratumoral CD8^+ T cell numbers independently of frOpn1 (Fig. 8 f).

## Discussion

Type I IFNs mediate successful immune responses against viral infections (McNab et al., 2015). High levels of IFN-α/β are produced by pDCs upon TLR7 and TLR9 activation by danger signals such as viral DNA and RNA (Blasius and Beutler, 2010; Gilliet et al., 2008; Kawai and Akira, 2011). At steady state, IFN-α and β are also constitutively weakly expressed, a process that prepares the cells to elicit quick and robust expression of IFN-α/β after activation of pattern recognition receptors (Honda and Taniguchi, 2006). Here we demonstrate that sOpn enhances this steady-state expression of IFN-β in pDCs, possibly promoting an alert mechanism against pathogenic antigens. In

agreement, exposure of Opn-primed pDCs to TLR triggering such as CpG-A results in enhanced type I IFN production. Importantly, this novel immune mechanism finds its application as an emerging immunotherapy factor capable of controlling tumor growth.

Under endotoxin-free conditions and in the absence of PAMPs, we demonstrate that ligation of α4-integrin by the SLAYGLR domain of Opn (frOpn1) enhances IFN-β expression in pDCs. In carcinogenesis, pDCs are present in high numbers in different tumors upon activation with TLR agonists (Liu et al., 2008; Palamara et al., 2004; Sorrentino et al., 2010; Stary et al., 2007). Interestingly, it has been found that activated pDCs induce tumor regression through type I IFN production (Drobits et al., 2012). Also, natural human pDCs with an IFN signature were used for vaccination, as they induce antigen-specific T cell responses in melanoma patients (Tel et al., 2013). Type I IFN is one of the approved drugs for melanoma treatment (Zitvogel et al., 2015). This treatment induces tumor-repopulating cells to enter dormancy; also, high expression of IFN-β is correlated with tumor cell dormancy in melanoma patients (Liu et al., 2018). Accordingly, we show that intratumoral injection of frOpn1 mediates pDC recruitment and activation to express high levels of IFN-β restricting tumor growth. Depletion of pDCs reveals their critical role for tumor growth deterioration after frOpn1 activation. Opn is considered mostly tumorigenic (Chiodoni et al., 2010; Rittling and Chambers, 2004); nevertheless, our data show that in the case of melanoma, Opn present in tumors could be converted to an in situ anticancer agent. Thus, preexisting tumor Opn could be modified by thrombin to reveal its SLAYGLR domain, or a peptide containing this domain could be administered intratumorally for therapy.

Several studies have demonstrated that IFNs are able to act suppressively on malignant cells directly or indirectly by enhancing cytotoxic effects of pDCs, NK cells, and CD8^+ T cells (Drobits et al., 2012; Liu et al., 2008; Liu et al., 2018; Palamara et al., 2004; Sorrentino et al., 2010; Stary et al., 2007; Thyrell et al., 2002). Here, we show that Opn/SLAYGLR enhances IFN-β production by pDCs, as well as their responsiveness to TLR triggering in terms of enhanced IFN-α production. Also, early recruitment and IFN production of pDCs by Opn affected maintenance of IFN-γ–producing cytotoxic T cells, and these events resulted in tumor regression. As induction of type I IFNs by DCs is a cutting-edge therapy in melanoma patients (Kranz et al., 2016; Sabado et al., 2017; Tel et al., 2013), molecules optimizing type I IFN–mediated immune mechanisms are emerging factors for cancer immunotherapy.

For IFN-α/β production, an initial wave of *Ifnb* and *Ifna* gene transcription relies on IRF3 activation (Honda et al., 2006; McNab et al., 2015). This initial type I IFN production triggers the transcription of *IRF7*, which then mediates a positive feedback loop, leading to the induction of a second wave of gene transcription, including additional IFN-α–encoding genes (Honda et al., 2006; McNab et al., 2015). Consistent with this, we demonstrate that frOpn1 treatment of pDCs preferentially induces an initial wave of IFN-β production dependent on PI3K/Akt/mTOR/IRF3 pathway activation. This signaling cascade prepares pDCs to mount a robust IFN-α/β response,

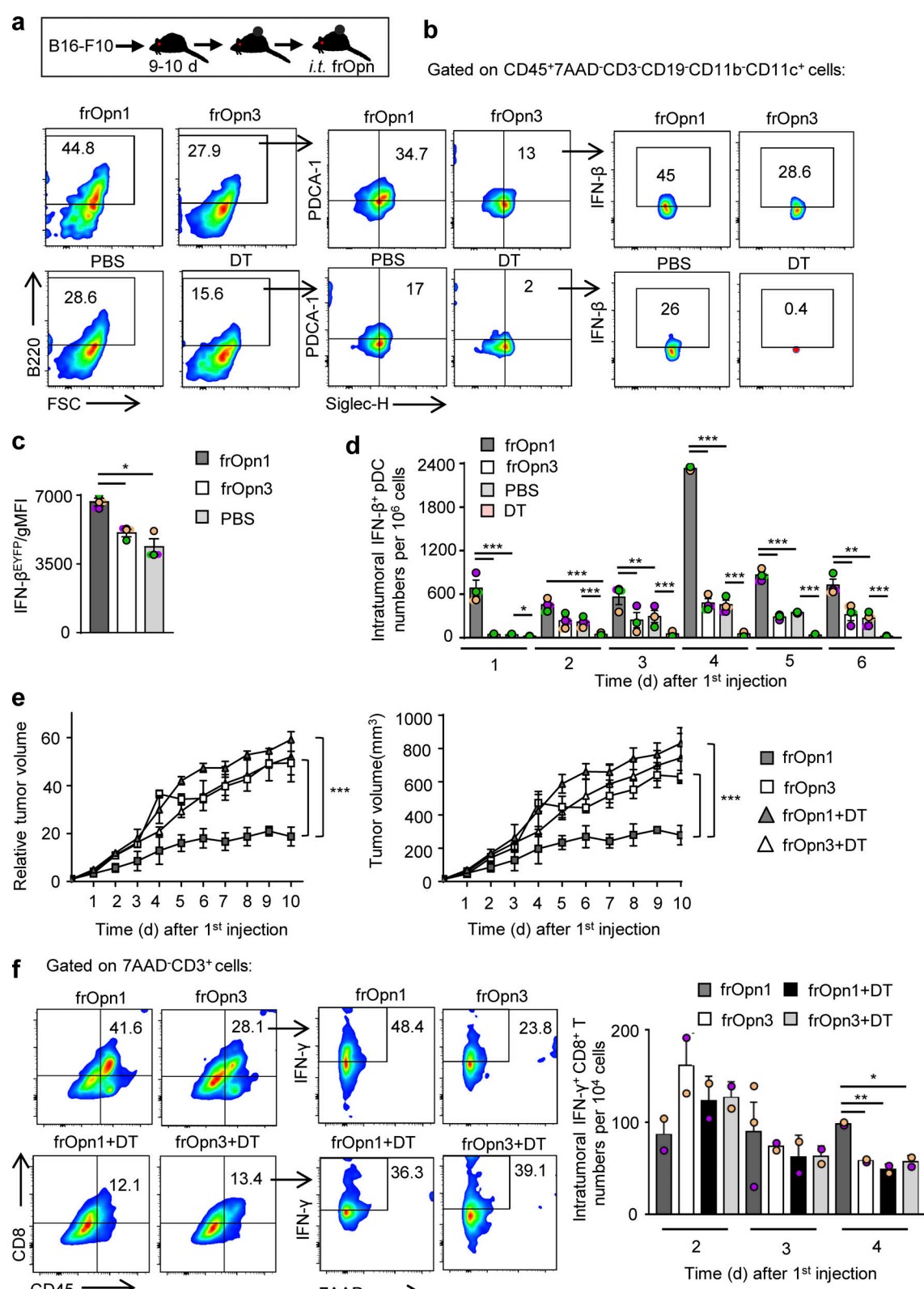

**Figure 8. Opn/SLAYGLR enhances the antitumor immune response through pDCs.** Mice presenting melanoma tumor after B16-F10 inoculation were treated with frOpn1/frOpn3 or PBS (control). **(a)** Experimental diagram of melanoma induction and treatment (i.t., intratumoral). **(b)** Percentages of pDCs (B220+PDCA1+Siglec-H+) gated from CD45+7AAD−CD3−CD19−CD11b−CD11c+ cells and IFN-β$^{EYFP}$ cells among this population present in B16-F10 melanomas induced in *Ifnb*$^{EYFP}$ mice on day 4 after the onset of daily frOpn injections or PBS. DT group demonstrates efficient pDC depletion in tumors of *BDCA2-DTR/ Ifnb*$^{EYFP}$ mice on day 4. FSC, forward scatter. **(c)** Geometrical mean fluorescence intensity (gMFI) of IFN-β$^{EYFP+}$-expressing pDCs. Data are presented as mean ± SEM (*n* = 3, pooled from 14 mice per group) from three independent experiments. **(d)** Kinetics of intratumoral IFN-β$^{EYFP}$ pDC numbers per 10$^6$ cells in tumors

of *Ifnb*[EYFP] and pDC-depleted *BDCA2-DTR/Ifnb*[EYFP] mice. Data are presented as mean ± SEM (*n* = 3, pooled from 14 mice per group) from three independent experiments. **(e)** Graphs showing the kinetics of tumor growth and tumor volume of melanomas induced in *BDCA2-DTR* ± DT mice, both measured at the indicated time points after the onset of frOpn injections. Data are presented as mean ± SEM (*n* = 3, pooled from five mice per group) from three independent experiments. In c and d, each experimental replicate is presented with different-colored dots, and dots with black line borders are the averages derived from each replicate. In c–e, *, P ≤ 0.033; **, P ≤ 0.002; ***, P ≤ 0.001 (two-way ANOVA with Bonferroni's multiple comparison test). **(f)** Percentages of CD8[+] T cells gated from CD45[+]7AAD[−]CD3[+] cells and IFN-γ[+] cells among this population present in melanomas of mice on day 4, and kinetics of intratumoral IFN-γ[+]CD8[+] T cell numbers per 10[4] cells. Data are presented as mean ± SEM (*n* = 2–3, pooled from five mice per group) from two independent experiments. *, P ≤ 0.033; **, P ≤ 0.002 (unpaired two-tailed Student's *t* test).

characterized by higher production of IFN-α upon activation by TLR ligation later on. This mechanism is dependent on a IFNAR1 positive feedback loop. In fact, the enhancement of IFN-α production by pretreatment with Opn/SLAYGLR following CpG-A stimulation depends on IFNAR signaling triggered by IFN-β. Thus, the integrin-binding SLAYGLR domain of Opn provided a low danger stimulus, increasing the baseline levels of IFN-β and enabling a second wave of robust production of IFN-α/β, after PAMP recognition. Whether this Opn/IFN-β alert prepares the immune system to distinguish trivial from pathogenic antigens merits investigation. This predisposition of pDCs caused by Opn may have an impact on certain autoimmune diseases connected to type I IFN expression levels (Ganguly et al., 2013).

Our previous studies revealed that the proinflammatory capacity of the gut CD103[−] DC subset depends on the interaction of Opn/SLAYGLR with α9 integrin (Kourepini et al., 2014). However, the possible effects of a specific integrin binding the SLAYGLR motif on the pDC subset function remained unknown. In this study, use of Opn synthetic fragments containing specific active motifs allowed the identification of α4β1 integrin as a receptor implicated in the production of IFN-β by pDCs. Apart from pDCs, α4 integrin is highly expressed in other innate cell populations as well. The role of Opn/Itga4 axis in cell populations such as monocytes, macrophages, neutrophils, and cDCs has been studied in various experimental settings (Kourepini et al., 2014; Lund et al., 2013; Uede, 2011). Previous findings, including studies of our group and others, indicate that the role of Opn/Itga4 axis in these myeloid populations is mainly on cell chemotaxis, recruitment, and survival (Kourepini et al., 2014; Lund et al., 2013; Uede, 2011). However, the induction of *Ifnb* expression by α4 integrin activation is a mechanism uniquely found in pDCs. More specifically, blockade of either α4-integrin or α4β1 suppressed Opn/SLAYGLR-mediated IFN-β production. On the other hand, blockade of α9 integrin in the presence of Opn/SLAYGLR treatment boosted significantly *Ifnb* expression in pDCs. In support of this phenomenon, upon α9 integrin blockade, we measured a threefold enhancement in *Itga4* mRNA expression in pDCs (not depicted). Considering that α9 and α4 integrins are the known receptors for Opn/SLAYGLR, our data suggest that α9 integrin blockade results in binding of frOpn1 to available and elevated α4 integrin, which further boosts *Ifnb* induction. It is therefore possible that integrins act as rheostats for IFN-α/β production. These findings may provide an additional explanation on the reactivation of dormant viruses in multiple sclerosis patients treated with natalizumab, which targets α4 integrin (Palmer, 2014). However, the conditions of α4 integrin activation in other settings remains to be elucidated, as this integrin interacts with several ligands and their specific

motifs (Uede, 2011), promoting a number of diverse biological functions (Gonzalez-Amaro et al., 2005).

In our setting, we specifically used the frOpn1 peptide for α4 integrin engagement, as it represents the thrombin-cleaved fragment of Opn with exposed cryptic SLAYGLR domain (Kourepini et al., 2014; Yokosaki et al., 1999). Thrombin cleaves Opn after the SLAYGLR aa sequence in two fragments, one containing the N-terminal and the other the C-terminal part. By using a 20 aa of the N-terminal part (icosamer), we restricted the reactive domains and avoided aa sequences that may contain sites capable of interacting with other molecules and possibly masking the effects of specific integrin binding. For example, next to the N-terminus, there is a calcium binding domain that could alter the effect of SLAYGLR domain of interest, as calcium has been found to suppress cell adhesion to Opn by attenuating binding affinity to integrins (Hu et al., 1995). In addition, located in the N-terminal part of Opn and next to the calcium site, there is a CD44 binding domain (Buback et al., 2009). Beside the N-terminal product, thrombin cleavage of Opn produces a fragment containing the C-terminal, which is also capable of interacting with several CD44 variants (Wang and Denhardt, 2008; Weber et al., 1996) and possibly affecting pDC function. We used the shortest active peptide version to dissect the role of SLAYGLR domain, which binds only to α4 and α9 integrins. In our previous studies, as well as in other settings, this icosamer was demonstrated to have greater efficiency compared with full-length or thrombin-cleaved Opn (Albertsson et al., 2014; Doyle et al., 2008; Kourepini et al., 2014). It is thus possible that interaction of Opn with other molecules (e.g., CD44), interferes with certain Opn effects such as pDC function. Of note, the icosamer is usually better tolerated upon therapeutic use in several disease models tested compared with larger fragments, or to the one that contains the complete N-terminal domain (Albertsson et al., 2014; Doyle et al., 2008; Jin et al., 2016; Zhou et al., 2020).

In pDCs, PI3K activation induces IFN-α/β production in response to TLR/MyD88 complex formation and activation of IRF7 (Cao et al., 2008). Here, we find that pDCs exposed to endotoxin-free SLAYGLR fragment leads to PI3K-mediated phosphorylation of Akt, independent of TLR-MyD88 complexes. It is known that Akt activation weakens the inhibitory effect of TSC1/2 through phosphorylation, leading to mTOR activation, while TSC1 deficiency also enhances its activity (Weichhart et al., 2015). In our setting, frOpn1 treatment boosted Tsc2 phosphorylation concomitantly to increment of Akt phosphorylation in pDCs. Nevertheless, recent studies in pDCs revealed that mTOR is activated not only by phosphorylation-mediated destabilization of TCS1/2 complex via Akt, but also by other factors that

limit *Tsc1* mRNA abundance (Agudo et al., 2014; Weichhart et al., 2015). As previously identified in pDCs, *Tsc1* transcript is a target of miR-126, leading to enhanced type I IFN production through mTOR activation (Agudo et al., 2014). Therefore, after Opn/SLAYGLR addition, we surveyed for possible alternations in the expression of genes associated with PI3K/mTOR pathway (*Akt* not depicted; *Tsc1* and *Tcs2*), and we found significant differences only in *Tsc1* mRNA expression. Moreover, use of *Akt*−/− mice and PI3K inhibitor Wortmannin revealed that the reduction in *Tsc1* mRNA observed in Opn/SLAYGLR-treated pDCs was operated via Akt. Whether the Opn-mediated suppression of *Tsc1* expression is a direct inhibitory effect of Akt or indirect through Akt's effect on specific mediators (e.g., miRNAs) still merits investigation. In our system, the integrin α4-PI3K-Akt pathway restrains both *Tsc1* mRNA and TCS1/2 complex by phosphorylation of Tsc2 in pDCs, mediating mTOR activation, demonstrated by increased phosphorylation of p70S6 kinase, an mTOR downstream molecule. Although activation of p70S6K is crucial for the TLR-mediated induction of IRF7 nuclear translocation and IFN-α/β production by pDCs (Cao et al., 2008), IRF7 activation was undetectable in our settings in the absence of autocrine IFN-α/β stimulation. Instead, IRF3 was overexpressed and activated, resulting in its nuclear translocation. Importantly, inhibition of several checkpoints involved in Opn activation of α4-integrin/PI3K/Akt/mTOR signaling network affected expression and nuclear translocation of IRF3 and production of IFN-β. IRF3 is constitutively expressed in mouse embryonic fibroblasts, inducing mainly IFN-β, whereas IRF7 activation induces both IFN-α and IFN-β (Honda et al., 2006; Sato et al., 2000). In agreement, we find that in pDCs, α4-integrin activation of PI3K by Opn in the absence of PAMPs activates mTOR/IRF3, leading primarily to IFN-β production. Given that several pathways downstream of PI3K signaling can induce low-level IFN-β production in the absence of strong IRF3/7 signaling (such as NF-κB), it was intriguing to know whether IRF3 is in fact the critical driver of IFN-β. In fact, IRF3-silenced pDCs exhibited diminished ability to enhance IFN-β levels after Opn/SLAYGLR treatment, implying that IRF3 is a crucial mediator of the Opn-induced IFN-β production observed via integrin α4.

Time course observations demonstrated that Opn-mediated enhanced *Irf3* transcription resulted in enhanced cytoplasmic IRF3 expression, followed by IRF3 phosphorylation/nuclear translocation. The significant boost in *Irf3* transcription (2.5–3 h) occurred concomitantly with the enhancement of cytoplasmic IRF3 production, indicating that newly translated IRF3 protein was translocated to the nucleus and not the preexisting cytoplasmic IRF3. In support, the timeframe for mTOR activation in our setting starts earlier than 1–1.5 h, as both Akt and Tsc2 phosphorylation took place within 30 min, and p70S6K phosphorylation started to exhibit statistically significant differences within 1 h after Opn/SLAYGLR addition. Therefore, we understand that IRF3 phosphorylation is not a result of a direct activating effect of mTOR, as the nuclear translocation of IRF3 takes place 2.5 h after Opn/SLAYGLR addition. For a direct posttranslational activating effect of mTOR, we would expect a swift response at a time point earlier than 2.5 h. The above data suggest that the activation of PI3K/Akt/mTOR increases baseline values of IRF3 by upregulated transcription, and it is not activated by mTOR directly at a posttranslational level.

In macrophages, nonpathogenic bacteria activate mTOR, promoting an immunoregulatory profile, while pathogenic bacteria induce suppression of mTOR function, promoting the expression of proinflammatory cytokines (Ivanov and Roy, 2013). It appears that mTOR signaling, in the presence of low danger signals, acts as a regulator for an initial wave of IFN-β production, and this effect is capable of tilting immune response toward antiinflammatory settings (Ivanov and Roy, 2013). Our data show that sOpn and its receptor α4β1 integrin can act as low danger signals, regulating initial or steady-state IFN-β production. It is possible that levels of Opn expression in different organs and individuals may affect basal levels of IFN-β (Chiocchetti et al., 2005; Chiocchetti et al., 2004; Gazal et al., 2015). In addition, different Opn haplotypes affecting its expression may impact IFN-β levels, influencing homeostasis and disease (Chiocchetti et al., 2005; Chiocchetti et al., 2004; Gazal et al., 2015; Giacopelli et al., 2004). Future investigation will shed light on these aspects.

It was unexpected that Opn/SLAYGLR priming of pDCs would enhance the expression of IFN-β, independently of the adaptor molecules MyD88 and TRIF. This reveals a mechanism that bypasses TLR activation, since these adaptor molecules are indispensable for TLR downstream signaling (Yamamoto et al., 2002). In fact, even higher IFN-β expression and IRF3 nuclear translocation are observed in the absence of MyD88, which is consistent with a study showing that MyD88 inhibits IKKε-induced activation of IRF3 (Siednienko et al., 2011). Under TLR9 activation, IRF7-mediated induction of IFN-α expression in pDCs is dependent on the interaction of iOpn with MyD88 (Shinohara et al., 2006). Here, we find that IFN-β is induced in the absence of MyD88, curtailing the importance of the iOpn-MyD88 association for the steady-state expression of IFN-β in pDCs.

The type I IFN–Flt3L axis is important for the generation of pDCs from BM progenitor cells (Chen et al., 2013), and we also have obtained similar results on the role of type I IFNs on pDC development (not depicted). As Opn and SLAYGLR induce IFN-β in pDCs, we further evaluated its effect on in vitro Flt3L-driven development of pDCs. In this setting, Opn enhances pDC numbers, as well as their IFN-β expression, indicating a physiological effect of Opn on DC development. At the same time, cDCs decreased in numbers, showing that sOpn is an important factor influencing pDC/cDC ratios. In agreement, Flt3L cultures of *Spp-1*-deficient BM progenitors result in decreased pDC development compared with wild-type progenitors (not depicted). A recent study showed that in vivo–administered sOpn promotes lymphopoiesis (Kanayama et al., 2017); however, mechanistic explanation was not provided. We speculate that this process could be explained by the sOpn-mediated enhanced pDC development, as well as IFN-β expression. Future experiments will elucidate this.

In this report, we demonstrate that sOpn, via its integrin-binding SLAYGLR motif, activates α4-integrin, triggering a unique MyD88-independent PI3K/mTOR/IRF3 pathway that leads to enhanced IFN-β expression in pDCs. This newly

described integrin/IFN-β axis may be implicated in a wide array of immune responses where pDCs play instrumental role. Opn/SLAYGLR in pDCs not only enhanced their suppressive function against melanoma tumor growth in vivo, but also boosted Flt3L-driven pDC generation from BM progenitor cells. Overall, these findings generate new questions on the roles of osteopontin and integrins in both homeostatic and disease settings.

## Materials and methods

### Study design
The primary objective of this study was to define the importance of the integrin-binding SLAYGLR motif of sOpn, which induces IFN-β production in murine pDCs, in the absence of TLR activation. In all experiments, appropriate control groups were used, and mice were housed under the same environmental conditions and were age matched. Adult female mice were randomly placed in distinct experimental groups. Authors were blinded for cell counts and flow cytometric and tumor volume analysis. The number of mice in each group was determined by power calculations based on extensive previous experience with the model system and is defined in the respective figure legends. The number of independent replicates for each experiment is defined within the respective figure legends. No samples or animals were excluded from data analyses.

### Mice
C57BL/6J, B6.129-$Ifnb1^{tm1Lky}$/J ($Ifnb^{EYFP}$), B6.129S2-$Ifnar1^{tm1Agt}$/Mmjax ($Ifnar1^{-/-}$), C57BL/6J-Tg (Itgax-Cre,-EGFP) 4097Ach/J ($Itgax^{Cre}$), B6.129P2 (SJL)-Myd88$^{tm1Defr}$/J ($Myd88^{fl/fl}$), B6.129P2-$Akt1^{tm1Mbb}$/J ($Akt1^{-/-}$), B6.129P2-$Akt2^{tm1Mbb}$/J ($Akt2^{-/-}$), and B6-Tg(CLEC4C-HBEGF) 956Cln/J (designated $BDCA2$-$DTR$) mice were purchased from The Jackson Laboratory. $Myd88^{fl/fl}$ were crossed with $Itgax^{Cre}$ to generate DCs deficient in $Myd88$ ($Myd88^{-/-}$). Mice were housed at the Animal Facility of the Biomedical Research Foundation of the Academy of Athens. All mice in the animal facility were screened regularly with a health-monitoring program, in accordance with the Federation of European Laboratory Animal Science Association, and were free of pathogens. Experiments used sex- and age-matched mice aged 8–12 wk. During all experiments, mice were monitored daily. Littermates of the same genotype were randomly allocated to experimental groups. All procedures were in accordance with institutional guidelines and approved by the Institutional Committee of Protocol Evaluation together with the Directorate of Agriculture and Veterinary Policy.

### Generation, isolation, and in vitro conditioning of BM-derived pDCs
For generation of pDCs and cDCs, BM cells were extracted, and erythrocytes were removed by brief exposure to 0.168 M $NH_4Cl$. Cells were cultured at a density of $1.5 \times 10^6$ to $3 \times 10^6$/ml in RPMI-1640 with 10% (vol/vol) FBS with or without rhFlt3L (200 ng/ml) at 37°C in 10% $CO_2$ (Naik et al., 2007). On day 11, naive $7AAD^-CD3^-CD19^-CD11c^+CD11b^-B220^+PDCA1^+$Siglec-H$^+$ pDCs were sorted using a FACS Aria III flow cytometer (BD) to a purity ≥98% (Fig. S4 a) after enrichment with a CD11c MicroBead kit (Miltenyi Biotec). Sorted pDCs were cultured in the presence of 250 ng/ml rOpn (R&D Systems) or 18.2 ng/ml synthetic $Opn_{134-153}$ fragments (IVPTVDVPNGRGDSLAYGLR, frOpn1–3; Caslo Laboratory ApS) or PBS (Kourepini et al., 2014; Alissafi et al., 2018). The RGD domain (Arg-Gly-Asp) of frOpn is scrambled to RAA (Arg-Ala-Ala) in frOpn1. In frOpn2, the SLAYGLR (Ser-Leu-Ala-Tyr-Gly-Leu-Arg) domain is scrambled to LRAGLRS (Leu-Arg-Ala-Gly-Leu-Arg-Ser). The frOpn3 has both RGD and SLAYGLR scrambled, to RAA and LRAGLRS, respectively (Kourepini et al., 2014).

For blocking experiments, 10 µg/ml of either LEAF purified anti-mouse α4-integrin antibody (BioLegend) or a polyclonal anti-mouse α9-integrin antibody (R&D Systems), or the corresponding anti-mouse isotype control antibodies (BioLegend, R&D Systems), were added 0.5 h before addition of frOpn/PBS to pDCs. Both anti-integrin antibodies inhibit integrin binding to their ligands (Halvorson and Coligan, 1995; Kourepini et al., 2014). Detailed information on the antibodies used is shown in Table S1.

pDCs were also pretreated with 50 µM MyD88 inhibitor (Pepinh-MYD; InvivoGen), 50 µM TRIF inhibitor (Pepinh-TRIF; InvivoGen), or respective control peptide (Pepinh-control; InvivoGen), 100 nM Wortmannin (Calbiochem), 5 µM Akt1 inhibitor (Calbiochem), or 100 nM Rapamycin (Calbiochem) according to the manufacturer's instructions. MyD88 and TRIF inhibitors were tested and effectively inhibited type I IFN production by CpG-A-stimulated (ODN 1585 and control ODN 1585; InvivoGen) and poly (I:C)-stimulated (InvivoGen) pDCs, respectively, in Fig. 4, e and f. Isolated pDCs ($10^6$/ml) were pulsed (20 min) with 1 µg/ml CpG-A or poly(I:C) and cultured for 24 h. Where appropriate, pDC cultures were treated with CpG-A concomitantly with either 10 µg/ml of an ultra-purified IFN-β antibody (BioLegend) or the corresponding isotype control antibody (BioLegend). IFN-α/β was measured in supernatant cultures after 24 h of incubation. Extensive purification and sorting of pDCs resulted in $10^4$–$5 \times 10^5$ pDCs per milliliter of culture, and therefore high-sensitivity ELISA kits were used, as they can detect quantities in the range 7.8–500 pg/ml for IFN-β (Bio Legend) and 12.5–400 pg/ml for IFN-α (PBL Assay Science).

### In vivo experimental protocols
For tumor induction, B16-F10 melanoma cells of C57BL/6 background (H-2$^b$) were cultured in complete RPMI for 10 d. Orthotopic tumors (melanomas) were induced in waxed back skin of C57BL/6 mice by subcutaneous injection of $10^5$ B16-F10 melanoma cells (Drobits et al., 2012; Overwijk and Restifo, 2001). Tumors were injected with frOpn1 or frOpn3 (72 ng) daily after their formation (day 9–10 of induction until day of analysis). Tumors were measured with calipers by determining the greatest longitudinal and transverse diameters (length and width), and their volume was calculated by using the ellipsoidal formula, $\pi/6 \times$ (length × width)$^2$. The relative tumor volume for each time point represents the ratio between the measured volume and the volume at time 0 (start of frOpn injection). For pDC depletion in $BDCA2$-$DTR$ mice, 120 ng DT was administered intraperitoneally on days −4 and −3, followed by the frOpn treatment protocol on day 0 (Swiecki et al., 2010).

## Flow cytometry

Freshly isolated live draining lymph node cells (7AAD⁻; BD Biosciences) and in vitro BM-derived cells were stained with combinations of fluorochrome-conjugated antibodies against CD4, CD3, CD8α, CD45, CD11c, CD11b, B220, Siglec-H, PDCA-1, CD19, and NK1.1 (BioLegend, eBioscience, and BD Biosciences). For dead cell exclusion, cells were stained with 7AAD (BD Biosciences). For intracellular cytokine staining, cells were stained with surface markers and then with an antibody against IFN-γ (eBioscience). Detailed information on the antibodies used is shown in Table S2.

Intracellular cytokine expression was assessed by 25 ng/ml PMA (Sigma-Aldrich) and 1 µg/ml ionomycin calcium salt (Sigma) for a 5-h incubation, as well as with a Cytofix/Cytoperm Kit Plus (Golgiplug; BD Biosciences). To perform gating strategy and define positive populations, isotype control antibodies for all markers were used, and also unstained samples (BioLegend). In the gating strategy used in tumor-derived CD45⁺7AAD⁻CD3⁻CD19⁻CD11b⁻CD11c⁺ cells for pDC phenotyping, we used not only unstained total tumor cells, but also sorted IFN-β⁺ pDCs from IFN-β-YFP mice with melanoma for an extra verification of the positive populations (Fig. S4 b). Flow cytometric measurements were performed using Attune Flow Cytometer (Thermo Fisher Scientific) and cell sorting with FACS ARIAIII (BD). Analysis of data was performed with FlowJo (TreeStar).

## Quantitative RT-PCR analysis

Total RNA was extracted using Nucleospin RNA II Kit (Macherey-Nagel) from sorted 7AAD⁻CD3⁻CD19⁻CD11c⁺CD11b⁻B220⁺PDCA-1⁺Siglec-H⁺ pDCs previously enriched with CD11c⁺ MicroBeads (Miltenyi Biotec). For RNA quantification, the Quant-iT RNA Assay Kit (Invitrogen) was used. Up to 1 µg of RNA was used for each reaction of cDNA synthesis with SuperScript II reverse transcriptase (Invitrogen) and RiboLock RNase inhibitor (Thermo Fisher Scientific). Primers were designed (Eurofins MWG) using the Primer3 program. *Irf3* sense, 5′-CGTCTAGGCTGGTGGTTATT-3′, and antisense, 5′-TGTCCTTGCTTTCTTTGTGA-3′; *Irf7* sense, 5′-CCCTCAACACCCTAATACCT-3′, and antisense, 5′-ATAGCCAGTCTCCAAACAGC-3′; *Tsc1* sense, 5′-ATTGGAGAAGTGGGCAGATT-3′, and antisense, 5′-GGTATGGGAGAGAGGTTGGAG-3′; *Tsc2* sense, 5′-GGCTACACCCACCTATGAAA-3′, and antisense, 5′-ACCCCAAACAGACAAGACAA-3′; and *Ifnb* sense, 5′-CCTATGGAGATGACGGAGAA-3′, and antisense, 5′-TGGAGAGCAGTTGAGGACAT-3′. Real-time PCR was performed with SYBRGreen I (Molecular Probes) and Platinum Taq DNA polymerase (Invitrogen) in a StepOnePlus RT-PCR system (AppliedBiosystems). Analysis was performed using the ΔΔCt method, where Ct is threshold count. All values were normalized against expression of the housekeeping gene hypoxanthine phosphoribosyltransferase (*Hprt*; sense primer, 5′-GTGAACTGGAAAGCCAAA-3′, and antisense primer, 5′-GGACGCAGCAACTGACAT-3′).

## Immunoblot

Isolated pDCs were lysed in PhosSTOP cocktail inhibitor (Roche) and subjected to SDS-PAGE electrophoresis on 7.5% gels and transferred to Immobilon-P^sq membrane (Millipore). Membranes were blocked with 5% skimmed milk in Tris-buffered saline with 0.5% Tween 20 and incubated with anti-IRF3, anti-phospho-IRF3 (Ser 396), anti-Akt, anti-phospho-Akt (Ser473), anti-phospho-TSC2 (Thr1462), anti-phospho-p70S6K (Ser371), and anti-phospho-p70S6K (Thr389), used at 1:1,000 (Cell Signaling), or β-actin (D6A8; 1:2,000; Cell Signaling) as control. Detection was performed using HRP-linked antibodies (Cell Signaling) and SuperSignal West Pico (Thermo Fisher Scientific). Detailed information on the antibodies used is described in Table S3.

## Immunofluorescence staining

Sorted 10⁵ pDCs were seeded in coverslips pretreated with poly-L-lysine and fixed with 4% PFA for 15 min at room temperature, followed by 10 min of fixation with ice-cold methanol. Cells were permeabilized by using 0.1% saponin (Sigma-Aldrich) in 2% BSA/PBS (Sigma-Aldrich; PS buffer) and stained with a rabbit anti-mouse IRF3 antibody (1:50; Abcam), followed by incubation with an Alexa Fluor 647–conjugated anti-rabbit IgG antibody (1:200; Life Technologies). Detailed information on the antibodies used is described in Table S3. For visualization of the nuclei, DAPI (Sigma-Aldrich) was used. Coverslips were mounted with ProLong Gold Antifade Mountant (P10144; Thermo Fisher Scientific), and images were captured using a scanning inverted confocal live cell imaging system Leica TCS SP5 (Leica Microsystems) with a 63×/1.4-NA oil-immersion lens and Leica Application Suite AF software (Leica Microsystems). The images were acquired at room temperature in sequential steps using the following settings: 8-bit acquisition, line averaging of three, a pinhole of 2 airy units, and scan speed 400. Image processing by Fiji software included 2D projections of z-stacks that were generated based on maximum intensities, colocalization analysis, fluorescence intensity measurements in regions of interest, and line profiling.

## Gene silencing

IRF3 gene expression was knocked down in pDCs by an IRF3 siRNA (Santa Cruz Biotechnology). IRF-3 siRNA (m) contained a pool of three different siRNA duplexes: A sense, 5′-GUUGUUCCUACAUGUCUUATT-3′, and antisense: 5′-UAAGACAUGUAGGAACAACTT-3′; B sense, 5′-CCAACUCUUUCCUCCUGAATT-3′, and antisense: 5′-UUCAGGAGGAAAGAGUUGGTT-3′; and C sense, 5′-CAACUCUUUCCUCCUGAAATT-3′, and antisense: 5′-UUUCAGGAGGAAAGAGUUGTT-3′. A control siRNA (si-Ctrl) was also used, containing a scrambled sequence that does not lead to the specific degradation of any known cellular mRNA, previously used in mouse pDCs: 5′-UUCUCCGAACGUGUCACGUTT-3′ (Eurofins MWG; Ko et al., 2018). For the most efficient siRNA delivery into pDCs and prevention of possible induction of cell death or pDC activation, we used a method based on the lipid-based reagent DOTAP (Roche Applied Sciences), as previously described for human pDCs (Smith et al., 2016). In detail, a volume of siRNA for a final concentration of 160 nM was diluted in PBS (1:5), and DOTAP was added (vol/vol). The mix was incubated at room temperature for 15 min and added to 10⁵ pDCs/100 µl culture (37°C, 10% CO₂ incubation).

## Statistics

Data were analyzed using unpaired two-sided Student's *t* test for statistical analyses of two-group comparisons. Multigroup comparisons were performed using two-way ANOVA, followed by the Bonferroni correction for the multiplicity of tests. Results are presented as mean ± SEM. P values <0.05 were considered statistically significant. Actual P values and number of replicates (*n*) are reported in each figure legend. Compared samples were collected and analyzed under the same conditions, and no data were excluded. In both types of parametric tests, data distribution was assumed to be normal by an *F* test of unequal variance. To show variability in the results between the different experiments and at the same time demonstrate averages of all the replicates, several data were presented in dot plots described as "superplots" (Lord et al., 2020). In these plots, each independent experiment is presented with a different color, and the averages derived from replicates are shown with black-bordered dots (Lord et al., 2020). Statistical analysis was performed between the mean values of each experiment (Lord et al., 2020). All statistical analyses were performed in Prism 7 software (GraphPad).

## Online supplemental material

Fig. S1 a shows the effect of SLAYGLR motif of Opn on mean fluorescence intensity (MFI) of IFN-β on BM-derived pDCs from B6.129-*Ifnb*<sup>EYFP</sup> mice; Fig. S1 b shows the effects and the efficiency of pDCs transfected with si-IRF3 or si-Ctrl. Fig. S2 is an *Tsc2* expression graph of frOpn-conditioned *Akt1*<sup>−/−</sup> and *Akt2*<sup>−/−</sup> pDCs. Fig. S3 depicts Opn and pDC depletion–mediated changes in different tumor-infiltrating leukocyte subsets. Fig. S4 depicts the gating strategy used for pDC sorting from Flt3L BM cultures (a) and for tumor-derived pDCs (b). Tables S1, S2, and S3 describe clone, species reactivity, source, conjugation, manufacturer, and catalog numbers of all antibodies used in this study.

## Acknowledgments

This paper is dedicated to the memory of our beloved professor V. Panoutsakopoulou (1967–2018). The authors thank M. Bessa for assisting with manuscript editing, A. Apostolidou for flow-cytometric sorting of cellular populations, E. Rigana for imaging acquisition and processing, and A. Koniaris for assistance with software handling. The authors also thank Amgen and Celldex Therapeutics for kindly providing hFlt3L.

The research leading to these results has received funding from the European Research Council under the European Union's Seventh Framework Program (FP7/2007-2013)/European Research Council Grant Agreement no. (243322; V. Panoutsakopoulou). E. Kourepini and N. Paschalidis are the recipients of State Scholarship Foundation postdoctoral scholarships (2017–2019).

The authors declare no competing financial interests.

Author contributions: DCM Simoes: designed and performed experiments, wrote the manuscript, and analyzed data. N. Paschalidis: designed and performed experiments, analyzed data, and performed statistical analysis. E. Kourepini: designed and performed experiments, analyzed data, wrote the original and the revised manuscript, performed statistical analysis, and supervised the study. V. Panoutsakopoulou: wrote the original draft and supervised the study.

Submitted: 9 February 2021

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

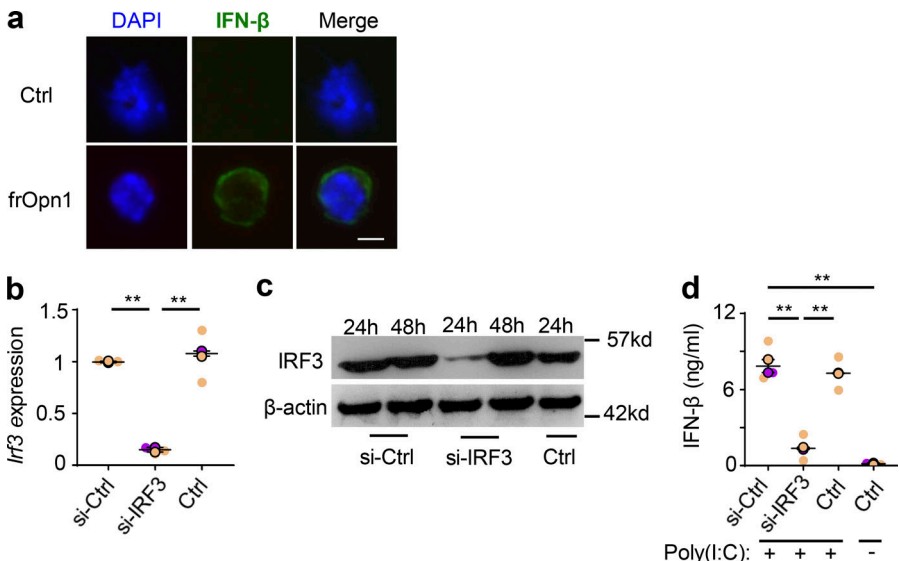

**Figure S1.** **The SLAYGLR motif of Opn induces IFN-β production by pDCs, and IRF3 is directly necessary for Opn/SLAYGLR-mediated IFN-β induction.** BM-derived pDCs from *B6.129-Ifnb*^EYFP mice treated with frOpn1 or PBS (control). **(a)** Representative images depicting IFN-β nuclear MFI and DAPI (blue). Scale bar, 5 µm. pDCs were transfected with siRNA targeting IRF3 (si-IRF3) or si-control (si-Ctrl). Ctrl are nontransfected pDCs. **(b and c)** *Irf3* expression normalized to *hprt* after 24 h (b) and immunoblotting after 24 and 48 h of culture with frOpn1 (c). **(d)** Levels of IFN-β measured in supernatants of pDCs treated with poly(I:C) TLR3 agonist after transfection with siRNA and in nontransfected controls. Data are presented as mean ± SEM (*n* = 3 wells per group) from two independent experiments. Each experimental replicate is presented with different-colored dots, and dots with black line borders are the averages derived from each replicate. **, P ≤ 0.002 (two-way ANOVA with Bonferroni's multiple comparison test).

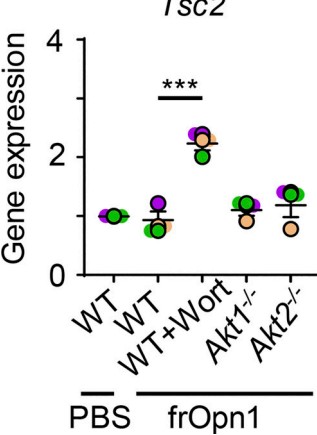

**Figure S2.** **Expression of *Tsc2* in frOpn-conditioned pDCs.** BM-derived pDCs from WT, Akt1^−/−, and Akt2^−/− mice were treated in vitro with frOpn1 or PBS (control) and pretreated with or without Wortmannin. Data are mean ± SEM (*n* = 3 wells per group) from three independent experiments. Each experimental replicate is presented with different-colored dots, and dots with black line borders are the averages derived from each replicate. ***, P ≤ 0.001 (two-way ANOVA with Bonferroni's multiple comparison test).

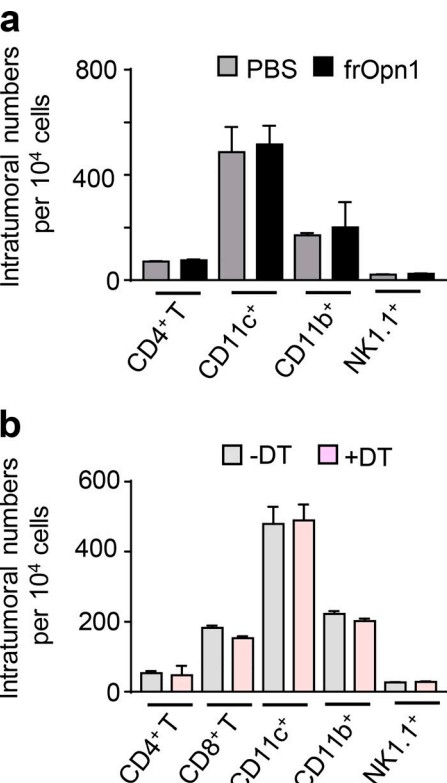

Figure S3. **Opn/SLAYGLR and pDC depletion–mediated changes in tumor-infiltrating leukocyte subsets. (a)** Graphs of CD4[+] T cells, cDCs, macrophages, and NK cell numbers in tumors on day 4 after injection of frOpn1 (black bars) or PBS (control, gray bars). **(b)** Graphs of CD4[+] T cells, CD8[+] T cells, cDCs, macrophages, and NK cell numbers in tumors on day 8 after first DT administration in *BDCA2-DTR* mice (day 4 after frOpn1 injection). Data are mean ± SEM (*n* = 2–3 wells per group) from two independent experiments. *, P ≤ 0.033; **, P ≤ 0.002; ***, P ≤ 0.001 (two-way ANOVA with Bonferroni's multiple comparison test).

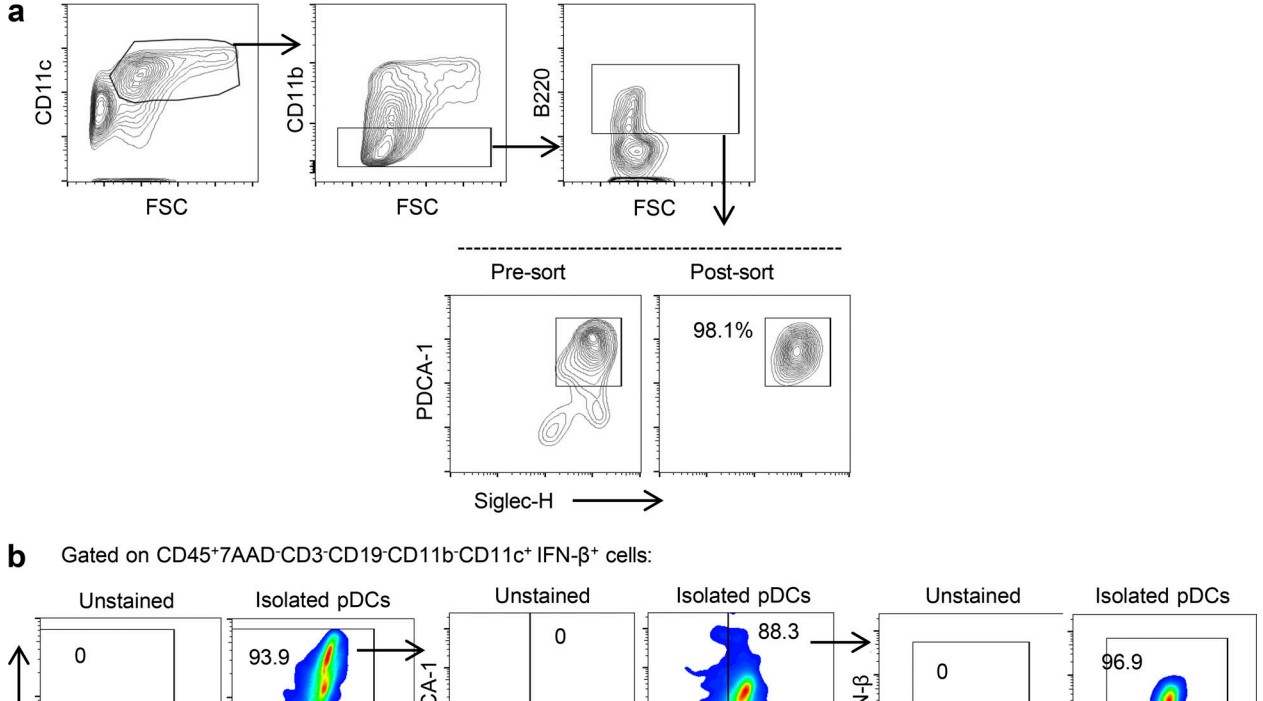

Figure S4. **Gating strategy for BM-derived and tumor-derived pDCs. (a)** 7AAD⁻CD3⁻CD19⁻CD11c⁺CD11b⁻B220⁺PDCA1⁺Siglec-H⁺ BM-derived pDCs were sorted to purity ≥98%. FSC, forward scatter. **(b)** Gating strategy used in tumor-derived CD45⁺7AAD⁻CD3⁻CD19⁻CD11b⁻CD11c⁺ cells for pDC phenotyping. Left: Unstained total tumor cells. Right: Sorted IFN-β⁺ pDCs from IFN-β^YFP mice with melanoma.

**Provided online are Table S1, Table S2, and Table S3. Table S1 shows information on antibodies used in blockade/neutralization and corresponding isotype controls. Table S2 shows information on fluorochrome-conjugated antibodies used in FACS. Table S3 shows information on antibodies used for immunofluorescence and immunoblotting.**

