## [Peer Review File · The Journal of Cell Biology]

An integrin axis induces IFN- β production in plasmacytoid dendritic cells

Davina Simoes, Nikolaos Paschalidis, Evangelia Kourepini, and Vily Panoutsakopoulou

Corresponding Author(s): Evangelia Kourepini, Biomedical Research Foundation of the Academy of Athens and Davina Simoes, Northumbria University

Review Timeline:

Submission Date:	2021-02-09
Editorial Decision:	2021-03-08
Revision Received:	2021-08-30
Editorial Decision:	2021-09-22
Revision Received:	2021-10-04

Monitoring Editor: Ira Mellman

Scientific Editor: Dan Simon

Transaction Report:

DOI: <https://doi.org/10.1083/jcb.202102055>

March 8, 2021

Re: JCB manuscript #202102055

Dr. Evangelia Kourepini
Biomedical Research Foundation of the Academy of Athens
Center for Basic Research
4 Soranou Efessiou Street
Athens 11527
Greece

Dear Dr. Kourepini,

Thank you for submitting your manuscript entitled "An integrin axis induces IFN- β production in plasmacytoid dendritic cells". Your manuscript has been assessed by expert reviewers, whose comments are appended below. Although the reviewers express potential interest in this work, significant concerns unfortunately preclude publication of the current version of the manuscript in JCB.

You will see that the reviewers were not yet convinced by your model and asked for additional investigations of the TRIF/MyD88-independent integrin-IRF3 axis. In particular, they shared several valid concerns related to the activation of IRF3 and requested additional controls throughout the study. Their points are constructive and we agree with the referees that these analyses would be important to bolster the conclusions.

Please let us know if you are able to address the major issues raised by the reviewers and wish to submit a revised manuscript to JCB. Note that a substantial amount of additional experimental data likely would be needed to satisfactorily address the concerns of the reviewers. As you may know, the typical timeframe for revisions is three to four months. However, we at JCB realize that the implementation of social distancing and shelter in place measures that limit spread of COVID-19 also pose challenges to scientific researchers. Lab closures especially are preventing scientists from conducting experiments to further their research. Therefore, JCB has waived the revision time limit. We recommend that you reach out to the editors once your lab has reopened to decide on an appropriate time frame for resubmission. Please note that papers are generally considered through only one revision cycle, so any revised manuscript will likely be either accepted or rejected.

If you choose to revise and resubmit your manuscript, please also attend to the following editorial points. Please direct any editorial questions to the journal office.

GENERAL GUIDELINES:

Text limits: Character count is < 40,000, not including spaces. Count includes title page, abstract, introduction, results, discussion, acknowledgments, and figure legends. Count does not include materials and methods, references, tables, or supplemental legends.

Figures: Your manuscript may have up to 10 main text figures. To avoid delays in production, figures must be prepared according to the policies outlined in our Instructions to Authors, under Data Presentation, <https://jcb.rupress.org/site/misc/ifora.xhtml>. All figures in accepted manuscripts will be

screened prior to publication.

IMPORTANT: It is JCB policy that if requested, original data images must be made available. Failure to provide original images upon request will result in unavoidable delays in publication. Please ensure that you have access to all original microscopy and blot data images before submitting your revision.

Supplemental information: There are strict limits on the allowable amount of supplemental data. Your manuscript may have up to 5 supplemental figures. Up to 10 supplemental videos or flash animations are allowed. A summary of all supplemental material should appear at the end of the Materials and methods section.

If you choose to resubmit, please include a cover letter addressing the reviewers' comments point by point. Please also highlight all changes in the text of the manuscript.

Regardless of how you choose to proceed, we hope that the comments below will prove constructive as your work progresses. We would be happy to discuss them further once you've had a chance to consider the points raised. You can contact the journal office with any questions, cellbio@rockefeller.edu or call (212) 327-8588.

Thank you for thinking of JCB as an appropriate place to publish your work.

Sincerely,

Ira Mellman, Ph.D.
Editor, Journal of Cell Biology

Melina Casadio, Ph.D.
Senior Scientific Editor, Journal of Cell Biology

Reviewer #1 (Comments to the Authors (Required)):

In this study by Simoes et al, soluble osteopontin (Opn) is identified as a novel driver of low-grade interferon beta (IFN β) production by plasmacytoid dendritic cells (pDC). Mechanistically, the authors provide evidence that this occurs through triggering of PI3K/Akt/mTOR signaling downstream of alpha 4 integrin. This mechanism appears to be independent of MyD88 or TRIF signaling, potentially ruling out involvement of TLR9 and several other pattern recognition pathways (notably, intracellular Opn was shown previously to support TLR9-driven IFN alpha production, a seemingly distinct pathway from what is described in the present manuscript; Shinohara, Nat Immunol, 2006). The authors propose that Opn-induced PI3K signaling ultimately converges on IRF3 to promote IFN β production. Finally, the authors demonstrate that intratumoral Opn injection can potentially promote anti-tumor immunity in mice through a mechanism dependent on pDC.

Overall, this study is intriguing and offers new insights into the biological roles of Opn, and how low-grade IFN β production can be regulated by non-TLR-based signaling axes. Given that Opn can be produced at high levels by activated T cells, it is intriguing to speculate that T cell-derived Opn might regulate type I interferon production in the context of anti-tumor immunity. However, I feel that the impact and interpretability of this study is limited by several factors (including lack of

experimental controls and logical gaps) that should be addressed prior to acceptance for publication. My specific comments are as follows:

Major concerns:

1. Fig. 3B/C: panel B shows a western blot that appears to indicate an increase in pIRF3 but not total IRF3 in pDC after 3 hours of frOpn1 treatment. However, panel C appears to show a striking induction of total IRF3 in both cytoplasm and nucleus with the same treatment, which seems inconsistent with panel B. An increase in total IRF3 would nevertheless be consistent with the increased Irf3 mRNA expression noted in panel A. Can the authors reconcile this by providing total signal intensity measurements for IRF3 (cytoplasm + nuclear) from their confocal analysis? Alternatively, western blots for IRF3 in cytoplasmic versus nuclear fractions would also be appropriate. Is the WB data for IRF3 in panel B as good as it gets (the staining is rather weak with significant background for pIRF3)?
2. Fig. 3E: I assume the black bars indicate 3 hours of frOpn1 treatment, while the gray bars are untreated controls, but this is not specified. Why does Itga9 blockade cause induction of Ifnb? Is this at all related to frOpn1 or purely a function of Itga9 blockade? A control group treated with Itga9 blocking antibodies only (without frOpn1) is needed to help clarify this.
3. Fig. 4B: similar question to the one above-why does the MyD88 inhibitor induce IFN β production? Again, there is no control group with MyD88 inhibitor alone so one can't discern whether this is in any way related to frOpn1. Also, there does not seem to be any treatment group with control peptide (assuming Pepinh-Myd is being used to inhibit MyD88, but this isn't specified in Methods). Ultimately, if the purpose of these studies is to demonstrate that frOpn1 signaling does not require MyD88, the data from MyD88 wt vs ko cells seem more easily interpretable than those from the inhibitor-treated cells. Can the authors demonstrate that the MyD88 inhibitor (in their hands) blocks IFN production in response to known MyD88-dependent stimuli?
4. Fig. 4D: like the MyD88 inhibitor experiments, a negative control peptide is not included in these TRIF inhibition experiments, making it difficult to interpret the data. Are TRIF-deficient mice not available for this experiment? And like MyD88, can the authors demonstrate that the inhibitor blocks IFN production downstream of known TRIF-dependent stimuli?
5. Fig. 5C: The canonical PI3K-AKT-mTOR pathway is thought to operate through phosphorylation-mediated inhibition of TSC1/2 (driven by Akt), which liberates mTOR. Given that this is a post-translational process, why have the authors assessed Tsc1 mRNA expression?
6. Fig. 5A and D: No control groups are shown (either a t=0 timepoint, or more appropriately, timepoint-matched untreated control samples). Without untreated control samples, it is not clear whether the changes in pAKT and p70S6K are actually driven by frOpn1 (or if they're merely a product of culture time).
7. Fig. 7: how was gating performed for the flow cytometry analyses? Did the authors use isotype control antibodies for key markers or some other method to define positive populations? For T cell analysis, were cells evaluated for IFN γ expression after restimulation ex vivo, or were they stained directly after isolation of cells?
8. Fig. 7: are there any other Opn-induced changes in other leukocyte subsets such as CD4 T cells, classical DC, or macrophages? Does pDC depletion affect the immune infiltrate?

9. Fig. 7: what is the impact of pDC depletion in the absence of frOpn1 treatment? Depletion of pDC clearly increases tumor growth in frOpn1 mice, but whether this is also true in mice treated with the inactive frOpn3 peptide is not explored. Without an frOpn3 + DT group, it is difficult to formally claim that pDC are actually required for the therapeutic effect of frOpn1, (ie, pDC depletion may affect tumor growth independently of frOpn1). It would also be useful to provide data indicating the efficacy of pDC depletion in DT-treated mice.

10. The authors propose a mechanistic model by which Opn/Itga4 activates PI3K/Akt/mTOR signaling, which in turn results in increased IRF3 activation and subsequent IFN β production. However, while their graphical abstract suggests a direct post-translational activating effect of mTOR on IRF3, their data in fact suggest that PI3K/mTOR signaling somehow promotes Irf3 transcription in response to Opn (Fig. 5F, and also Fig. 3A). This seems to be consistent with the apparent general increase in IRF3 protein level observed in their confocal experiments. To clarify this, I would urge the authors to carefully evaluate the levels of IRF3 mRNA, phospho-IRF3, and total IRF3 over a detailed time-course of several hours to better establish whether IRF3 is actually activated by PI3K/mTOR at a post-translational level, or if increased IRF3 activity is driven by upregulated transcription.

11. Mechanistically, the authors do not show through direct loss-of-function experiments that IRF3 is actually necessary for the induction of IFN β in response to Opn. Given that other pathways can induce low-level IFN β production in the absence of strong IRF3/7 signaling (such as NF κ B), the authors should consider approaches to establish whether IRF3 is in fact the critical driver of IFN β (e.g. using pDC from IRF3-deficient mice), or if it is another pathway downstream of PI3K signaling.

Minor concerns:

1. Fig. 3A: what are expression values normalized to ("relative" to what)?
2. Alpha 4 integrin (and beta 1 integrin) is expressed at high levels on other innate immune populations including monocytes, macrophages and conventional DC. Does Opn uniquely induce IFN β production in pDC, or does this mechanism also exist in other myeloid populations?
3. The IFNAR1 gene name is frequently misspelled as "IFNRA1" and should be corrected

Reviewer #2 (Comments to the Authors (Required)):

In this manuscript, the authors describe a novel axis that promotes a low level of IFN- β production by plasmacytoid dendritic cells in the absence of inflammatory signals. This axis is mediated by PI3K/mTOR signaling upon binding of the SLAYGLR motif of osteopontin to α 4 integrin. This is a potentially interesting finding with possible implications for pDC-mediated immune responses (especially anti-tumor), but there are several issues that must be addressed.

Major:

The peptides used by the authors for engagement of α 4 integrin are very short and it is not clear how strongly this binding resembles a ligation by actual products of thrombin-mediated osteopontin

cleavage. Also, Figure 5a and d lack multiple controls used in other experiments. Therefore, it is unclear to what extent the agonistic frOpn1 peptide triggers relevant signaling.

In Figure 3c, the authors quantify the nuclear/cytoplasmic ratio of IRF3 MFI. This should be done for all graphs depicting IRF3 MFI in order to better account for likely background expression in different cells/experiments.

In Figure 6, the authors claim that the enhancement of IFN- α production by pre-treatment with frOpn1 following CpG-A stimulation depends on IFNAR signaling triggered by IFN- β . However, the actual dependence of this effect on IFN- β is not directly addressed. Instead, only a complete dependence of any IFN- α production on the presence of IFN- α/β receptor is shown. The authors should block IFN- β in order to conclude that the "exaggerated responses to TLR triggering" depend on Opn/SLAYGLR-mediated IFN- β production.

Figure 7 lacks basic controls such as PBS. Also, only one group in Figure 7e has a DTR control. Moreover, such DTR controls are missing from crucial experiments showing effects on T cells in Figure 7f. In this Figure 7f, the authors show an increased number of intratumoral IFN- γ producing CD8+ T cells following frOpn1 treatment in comparison to frOpn3. However, this effect is not necessarily dependent on pDCs and could be possibly mediated by direct effects on T cells. Therefore, depletion of pDCs with DT (as in Figure 7e) is necessary for this conclusion. Further, the authors only show relative tumor volume 5 days after frOpn treatment (14-15 days total), which is a short time frame for the B16-F10 model. The authors should include the actual total tumor volume and follow the tumors to a later time point (such as day 25).

Minor:

Throughout the manuscript, the authors refer to the type-I interferon- α/β receptor gene as *lnra1*, but the proper gene name is *lnar1*. This should be corrected.

Line 168: Figure 8 is mistakenly called out.

Figure 3b: the representative immunoblot is not clear, which makes it difficult to interpret.

Figure 3d-e: Blockade of $\alpha 9$ integrin appears to increase IRF3 MFI, but it not shown to be statistically significant. Further, blockade of $\alpha 9$ integrin increases *lnb* expression. The authors discuss these results in lines 130-133 and only say that $\alpha 9$ integrin does not decrease IRF3 nuclear translocation or *lnb* expression. Can the authors comment on this?

Figure 5c: It is unclear what the statistics are comparing.

August 30, 2021

Dear Editors of the Journal of Cell Biology,

Thank you for considering a revised version of our manuscript [MS #202102055] which now addresses the reviewers' comments. We are thankful to the expert Editor for sending our work for review and to the reviewers for their careful, perceptive and positive comments. As you will read in our point-by-point response below, we have constructively addressed every point raised and have indicated corresponding changes in the revised manuscript (shown in yellow highlighting).

The reviewers' points have strengthened the manuscript as well as the proposed mechanisms. Of note, the comments were consistent between the reviewers, which helped us efficiently to fill in the logical gaps and the technical details. Among the amendments incorporated in the revised version, we clarified important details including description on the flow cytometric analyses for tumor-derived pDCs, data on the efficacy of pDC depletion in DT-treated mice, on the efficiency of MyD88/TRIF inhibition and on the role of $\alpha 9$ - integrin blockade in the enhancement of Opn-induced *Ifnb* expression by pDCs. By providing all the requested control groups, the representation of our findings was improved, clarifying the rationale for our conclusions. Comprehension of the proposed mechanism was substantially improved by adding several key descriptive points of our experimental setup and by furthering the discussion section.

We are thankful to the reviewers, as addressing their comments gave a new perspective to our findings. In summary:

We clarified our findings by concomitant evaluation of IRF3 mRNA, nuclear IRF3, and cytoplasmic IRF3 levels over a detailed time-course of several hours. This was an important aspect in our study, as now we can confidently propose that IRF3 was not directly activated by PI3K/mTOR at a post-translational level, and that increased IRF3 activity is driven by upregulated transcription. In addition, direct loss-of-function experimentation helped us, via IRF3 deletion, to verify that IRF3 is crucial for the Opn/SLAYGLR -induced IFN- β production observed through $\alpha 4$ -integrin. We also clarified that the induction of *Ifnb* expression by Opn/SLAYGLR -mediated $\alpha 4$ -integrin activation is a mechanism uniquely found in pDCs. Moreover, after IFN- β blockade in cultures, we now can conclude that the enhancement of IFN- α production by pre-treatment with Opn following CpG-A stimulation depends on IFN- β . The revised discussion on the utilized specific Opn fragments now clarifies the capacity of these fragments to effectively ligate $\alpha 4/\alpha 9$ -integrins. Particularly, the Opn/SLAYGLR icosamer frOpn1 (frOpn₁₃₄₋₁₅₃) is fully effective and resembles ligation by thrombin cleaved Opn. Moreover, our revised results on melanoma were highly enhanced, as now we demonstrate that pDC depletion does not affect tumor growth independently of Opn/SLAYGLR and we provide more detailed information up to the latest time point possible for the measurements of the tumor growth kinetics.

We apologize for the detailed and thus lengthy reply to the reviewers; however, we tried to be thorough. We sincerely hope you are satisfied that we have answered all of the reviewers' queries and find our revised manuscript suitable for publication in JCB.

With respect,

Evangelia Kourepini, Ph.D.

Corresponding author

ekourepini@bioacademy.gr

Point-by-point response

We thank the reviewers for carefully reading our manuscript as well as for their constructive critique.

Please, read our point-by-point response below.

Response to Reviewer #1:

(Reviewer's comments are in **bold** characters.)

Comments:

In this study by Simoes et al, soluble osteopontin (Opn) is identified as a novel driver of low-grade interferon beta (IFN β) production by plasmacytoid dendritic cells (pDC). Mechanistically, the authors provide evidence that this occurs through triggering of PI3K/Akt/mTOR signaling downstream of alpha 4 integrin. This mechanism appears to be independent of MyD88 or TRIF signaling, potentially ruling out involvement of TLR9 and several other pattern recognition pathways (notably, intracellular Opn was shown previously to support TLR9-driven IFN alpha production, a seemingly distinct pathway from what is described in the present manuscript; Shinohara, Nat Immunol, 2006). The authors propose that Opn-induced PI3K signaling ultimately converges on IRF3 to promote IFN β production. Finally, the authors demonstrate that intratumoral Opn injection can potentially promote anti-tumor immunity in mice through a mechanism dependent on pDC.

Overall, this study is intriguing and offers new insights into the biological roles of Opn, and how low-grade IFN β production can be regulated by non-TLR-based signaling axes. Given that Opn can be produced at high levels by activated T cells, it is intriguing to speculate that T cell-derived Opn might regulate type I interferon production in the context of anti-tumor immunity. However, I feel that the impact and interpretability of this study is limited by several factors (including lack of experimental controls and logical gaps) that should be addressed prior to acceptance for publication.

We sincerely thank the reviewer for thoroughly reviewing our manuscript as well as for the positive, explanatory comments and valuable suggestions which are now incorporated in the revised manuscript. Of note, most comments and suggestions were consistent among reviewers. We regret that relevant technical details were previously described briefly resulting in the impression that they were omitted. We have now added relevant important technical details in the Results and Methods sections. Below is a point-by-point reply to the comments.

My specific comments are as follows:

1. **Fig. 3B/C: panel B shows a western blot that appears to indicate an increase in pIRF3 but not total IRF3 in pDC after 3 hours of frOpn1 treatment. However, panel C appears to show a striking induction of total IRF3 in both cytoplasm and nucleus with the same treatment, which seems inconsistent with panel B. An increase in total IRF3 would nevertheless be consistent with the increased Irf3 mRNA expression noted in panel A. Can the authors reconcile this by providing total signal intensity measurements for IRF3 (cytoplasm + nuclear) from their confocal analysis? Alternatively, western blots for IRF3 in cytoplasmic versus nuclear fractions would also be appropriate. Is the WB data for IRF3 in panel B as good as it gets (the staining is rather weak with significant background for pIRF3)?**

We appreciate the reviewer's concern on the representation of this result. We reconciled the data by providing bar graphs showing western blot analysis for protein levels of total IRF-3 and p-IRF3, normalized to β -actin, which were both significantly increased upon frOpn1 addition when compared to the untreated controls (revised **Fig. 3b** and **Result section lines: 121-3**).

Graphs in Revised Figure 3b:

As suggested by the reviewer, the confocal analysis graph showing the ratio of nuclear/cytoplasmic IRF3 MFI was replaced by a dot plot demonstrating signal intensity of total (cytoplasmic + nuclear) IRF3 in individual cells, as well as nuclear IRF3 in comparison (revised Fig. 3c). In agreement to the graphs added in the revised Fig. 3b, total and nuclear IRF3 signal intensity were significantly enhanced upon frOpn1 addition (revised Fig. 3c and Result section lines: 123-5).

Plot in revised Figure 3c:

Moreover, for consistency of data presentation (as suggested also by the other reviewer-2nd comment-), we replaced all bar graphs showing MFI of nuclear IRF3 with dot plots demonstrating total and nuclear MFI in individual cells (revised Figures 3c, 3d, 4c and 5g).

Plot in revised Figure 3d:

Plot in revised Figure 4c:

Plot in revised Figure 5g:

For the pIRF3 western blot, an ultra-high sensitivity developing reagent was used (previously mentioned in method section line 409). Thus, in order to limit background appearance, we now provide a clearer/higher definition picture as requested by the reviewer (revised **Fig. 3b**).

WB in revised Figure 3b:

2. **Fig. 3E:** I assume the black bars indicate 3 hours of frOpn1 treatment, while the gray bars are untreated controls, but this is not specified. Why does Itga9 blockade cause induction of *Ifnb*? Is this at all related to frOpn1 or purely a function of Itga9 blockade? A control group treated with Itga9 blocking antibodies only (without frOpn1) is needed to help clarify this.

We thank the reviewer for these crucial comments. The meaning of the gray and black bars was previously specified on the top of Figure 3, but now we also added this information on the revised **Fig. 3e**, as well as in **Fig. legend 3e**. The legend also specifies the timeframe of treatment (3hrs).

We appreciate the importance of the question posed by the reviewer on whether $\alpha 9$ integrin blockade in pDCs induces *Ifnb* expression. The revised manuscript now includes the corresponding control groups of pDCs treated with either $\alpha 9$ integrin Ab or Ig Ab, without frOpn1 addition (PBS controls, gray bars, revised **Fig. 3e** right graph). By demonstrating these controls, we clarified that blockade of $\alpha 9$ integrin alone does not cause a significant increase in *Ifnb* expression in pDCs, but it is the presence of frOpn1 which is required for this increment. In support to this phenomenon observed, upon $\alpha 9$ integrin blockade we measured a 3-fold enhancement in *Itga4* mRNA expression in pDCs (data not shown). Considering that $\alpha 9$ and $\alpha 4$ integrins are the known receptors for frOpn1, our data suggest that $\alpha 9$ integrin blockade results in binding of frOpn1 to available and elevated $\alpha 4$ -integrin which further boosts *Ifnb* induction. We now discuss this point in the **Discussion section lines: 300-5**. These observations are vital for the comprehension of our results and answer also the other reviewer’s similar questions on $\alpha 9$ integrin blockade.

For consistency of data presentation, we included also the corresponding control groups for $\alpha 4$ integrin Ab and Ig Ab, without frOpn1 (PBS controls, gray bars, revised **Fig.3e** left graph).

Revised Figure 3e:

3. Fig. 4B: similar question to the one above-why does the MyD88 inhibitor induce IFN β production? Again, there is no control group with MyD88 inhibitor alone so one can't discern whether this is in any way related to frOpn1. Also, there does not seem to be any treatment group with control peptide (assuming Pepinh-Myd is being used to inhibit MyD88, but this isn't specified in Methods). Ultimately, if the purpose of these studies is to demonstrate that frOpn1 signaling does not require MyD88, the data from MyD88 wt vs ko cells seem more easily interpretable than those from the inhibitor-treated cells. Can the authors demonstrate that the MyD88 inhibitor (in their hands) blocks IFN production in response to known MyD88-dependent stimuli?

Reviewer's comments for Figure 4b were also to the point and very helpful. In order for our data to be more interpretable (such as the data from *MyD88*^{-/-} cells in Fig. 4a), we added a control group of pDCs treated with MyD88 inhibitor alone (PBS control, gray bar, revised Fig. 4b and Result section line: 148). As requested by the reviewer, we added also the control peptide of MyD88 inhibitor for both PBS- and frOpn1-treated pDC groups, (Ctrl Inhibitor, revised Fig. 4b and Result section line: 148-9).

Utilization of Pepinh-MYD to inhibit MyD88 (MyD88 inhibitor) and of Pepinh-control as the specific control peptide of MyD88 inhibitor (Ctrl Inhibitor), are now specified (revised Fig. legend 4b and Method section lines: 453-5). Thus, it is observed that frOpn1 treatment of *Ifnar1*^{-/-} pDCs enhances IFN- β levels in the presence of MyD88 inhibitor, which is in agreement with Figure 4a where *MyD88*^{-/-} cells were used. Therefore, frOpn1 induces IFN- β without the requirement of MyD88 signaling pathway.

Revised Figure 4b:

As mentioned in the original version of the manuscript, the functionality of MyD88 inhibitor was tested (previously on Materials and methods lines 372-3). We now demonstrate that treatment with MyD88 inhibitor (Pepinh-MYD, Invivogen), effectively blocked IFN- α production by wild-type pDCs in response to CpG-A oligonucleotide, which is a TLR9 agonist and a known MyD88-dependent stimuli (new Fig. 4e and Method section lines: 457-9).

Figure 4e:

4. Fig. 4D: like the MyD88 inhibitor experiments, a negative control peptide is not included in these TRIF inhibition experiments, making it difficult to interpret the data. Are TRIF-deficient mice not available for this experiment? And like MyD88, can the authors demonstrate that the inhibitor blocks IFN production downstream of known TRIF-dependent stimuli?

As fairly suggested by the reviewer, we added the control group with TRIF inhibitor alone (PBS control, gray bar, revised Fig. 4d), as well as the control peptide of TRIF inhibitor (Ctrl Inhibitor, revised Fig. 4d). Like *MyD88*^{-/-}, the *TRIF*^{-/-} mice were difficult to breed and grow. In contrast to *MyD88*^{-/-} mice, unfortunately we were unable to have a substantial number of *TRIF*^{-/-} littermates in order to isolate the appropriate amount of pDCs for the experiment suggested. The authors opted for transient inhibition (TRIF inhibitor) for the most reliable solution in our setting, as possible compensatory mechanisms could be induced in the TRIF-deficient mice altering the signaling pathways involved in type I IFN production. Therefore, in addition to demonstrating IFN-β secretion levels, we now also exhibit measurements of *ifnb* mRNA expression in the same control groups mentioned above (revised Fig. 4d left graph).

The use of Pepinh-TRIF to inhibit TRIF (TRIF inhibitor) and of Pepinh-control (Ctrl Inhibitor) as the specific control peptide of TRIF inhibitor, are now specified (Fig. legend 4d and Method section lines: 453-5).

Revised Figure 4d:

As mentioned in the original version of the manuscript, the functionality of TRIF inhibitor was tested (previously on Materials and methods lines 372-3). We now provide a demonstration where the TRIF inhibitor utilized (Pepinh-TRIF, Invivogen), efficiently blocked IFN-β production by pDCs in response to poly (I:C) that is a TLR3 agonist and a known TRIF-dependent stimuli (new Fig. 4f and Method section lines: 457-9).

Figure 4f:

5. Fig. 5C: The canonical PI3K-AKT-mTOR pathway is thought to operate through phosphorylation-mediated inhibition of TSC1/2 (driven by Akt), which liberates mTOR. Given that this is a post-translational process, why have the authors assessed Tsc1 mRNA expression?

We thank the reviewer for this important comment which helped us to clarify the purpose of *Tsc1* mRNA expression measurement in the revised manuscript. The original manuscript version only mentions that TSC1 deficiency enhances mTOR activity (Weichhart *et al. Nat Rev Immunol*, 2015, lines 272-273). The reviewer correctly indicates that the canonical PI3K-AKT-mTOR pathway operates through a post-translational process where Akt drives the phosphorylation-mediated inhibition of TSC1/2 liberating mTOR. However, recent studies in pDCs revealed that mTOR is also activated by factors that limit *Tsc1* mRNA abundance (Agudo *et al. Nat Immunol* 2014, Weichhart *et al. Nat Rev Immunol*, 2015). Agudo *et al.* demonstrated that miR-126 in pDCs targets *Tsc1* transcript and its decrease leads to enhanced type I IFN production through mTOR activation. In our setting, we looked for possible frOpn1-mediated alternations in the expression of genes associated with PI3K/mTOR pathway (*Akt*, *Tsc1* and *Tcs2*) and we found significant differences only in *Tsc1* mRNA expression (Fig. 5c, new Fig. S2 and data not shown). In addition, utilization of *Akt*^{-/-} mice and PI3K inhibitor Wortmannin (Fig. 5c) reveal that the reduction in *Tsc1* mRNA observed in frOpn1-treated pDCs was operated via Akt. Subsequently, *Akt*^{-/-} and Wortmannin frOpn1-treated pDCs had limited production of IFN-β when compared to controls (Fig. 5b). Whether the Opn-mediated suppression of *Tsc1* expression is either a direct inhibitory effect of Akt, or indirect through Akt effect on specific mediators (e.g. miRNAs), still merits investigation. We now added these observations and explanation in the revised manuscript (Result section lines: 163-5 and Discussion section lines: 336-46).

In order to investigate the possible post-translational process of TSC1/2 inhibition, we measured phosphorylated Tsc2 protein in PBS- and frOpn-treated pDC groups. We added a western blot image and a quantification bar graph, demonstrating significantly raised Tsc2 protein phosphorylation after frOpn1-treatment of pDCs, with highest levels at 30 min which coincides with the time point where Akt is also highly phosphorylated (revised Fig. 5a). We now add these data in Result section lines: 161-3, Method section line: 504 and Discussion section lines: 335-6, 346-9.

Revised Figure 5a:

6. Fig. 5A and D: No control groups are shown (either a t=0 timepoint, or more appropriately, timepoint-matched untreated control samples). Without untreated control samples, it is not clear whether the changes in pAKT and p70S6K are actually driven by frOpn1 (or if they're merely a product of culture time).

As the reviewer noticed, without the demonstration of untreated control samples in Figures 5a and d, it was not so clear whether the changes in pAkt and p70S6K were driven by frOpn1. Previously, we had mentioned the use of control groups only in the methods section (original manuscript lines 410-1). Thus, we now included the timepoint matched control samples (PBS-treated pDCs) as the most appropriate control groups in both revised Figures 5a (please see the figure in our answer to the 5th comment) and 5d. We also added matched quantitative graphical plots for each protein demonstrated in western blots, normalized to β-actin (revised Fig. 5a, 5d). Utilization of the control pDCs verified that phosphorylation of Akt was frOpn1-driven (revised Fig. 5a). Moreover, p70S6 kinase was indeed phosphorylated through culture time, whereas frOpn1 treatment further

enhanced significantly the amounts of both phosphorylated Ser371 and Thr389 when compared to the respective time-points of the control group (revised **Figure 5d**). We added these results in the revised **Result section lines: 158-9, 170-3**. Consistently, this suggestion is also in fully agreement with the other reviewer's 1st comment.

Revised Figure 5d:

7. **Fig. 7: how was gating performed for the flow cytometry analyses? Did the authors use isotype control antibodies for key markers or some other method to define positive populations? For T cell analysis, were cells evaluated for IFNγ expression after restimulation *ex vivo*, or were they stained directly after isolation of cells?**

We thank the reviewer for this comment which gave us the opportunity to better describe the flow cytometric analyses used specifically for tumor-derived pDCs. In order to define positive populations, the strategy for analysis included comparison to isotype control antibodies for all studied markers. To further verify our gating strategy for the positive/negative populations, we used unstained samples, as well as also sorted IFN-β⁺ melanoma-derived pDCs (**new Fig. S4b**). The revised manuscript includes the above information also in **Method section lines 483-87**.

Figure S4b:

For intracellular cytokine analysis, T cells were evaluated for IFN-γ expression after a 5h *ex vivo* re-stimulation (PMA/Ionomycin/GolgiPlug). This information is now included in the revised **Method section lines 479-83**.

8. Fig. 7: are there any other Opn-induced changes in other leukocyte subsets such as CD4 T cells, classical DC, or macrophages? Does pDC depletion affect the immune infiltrate?

We thank the reviewer for this important point, as CD4⁺ T cells, cDCs and macrophages express Opn receptors and thus could be affected. While carrying out the preliminary studies on melanoma, we studied the recruitment of differential leukocyte subsets. We observed Opn-induced changes in intratumoral pDCs and CD8⁺ T cell numbers, as described in **Figure 7**). In contrast, the recruitment/cellularity of CD4⁺ T cells, cDCs, macrophages and NK cells were not significantly altered on day 4 after frOpn1 injection. We now provide a graph with numbers of these tumor infiltrating leukocyte subsets on day 4 after frOpn1 injection in the **new Fig. S3a**. Measurements are also mentioned in the **Result section lines: 233-4**.

Figure S3a:

Whether pDC depletion affects the immune infiltrate is also an intriguing question that needed to be explored in our setting. Therefore, while depleting pDCs, we analyze the recruitment and numbers of the immune cells infiltrated in tumors. Apart from the pDCs which were depleted upon DT administration (revised **Fig. 7b, 7d**), there were not any significant changes in the numbers other immune cell subsets. In the revised manuscript we provided a graph with different leukocyte subsets on day 8 after first ±DT administration (day 4 after frOpn1 injection) in the **new Figure S3b** and we mention our findings in the **Result section lines: 222-3**.

Figure S3b:

9. Fig. 7: what is the impact of pDC depletion in the absence of frOpn1 treatment? Depletion of pDC clearly increases tumor growth in frOpn1 mice, but whether this is also true in mice treated with the inactive frOpn3 peptide is not explored. Without an frOpn3 + DT group, it is difficult to formally claim that pDC are actually required for the therapeutic effect of frOpn1, (ie, pDC depletion may affect tumor growth independently of frOpn1). It would also be useful to provide data indicating the efficacy of pDC depletion in DT-treated mice.

We are grateful to the reviewer for raising this crucial question and giving us the opportunity to clarify our findings. The impact of pDC depletion in the absence of frOpn1 treatment is now analyzed and it is presented in the revised manuscript: In **Figure 7e** where the relative tumor volume is demonstrated, we added a group of pDC-depleted mice treated with the inactive peptide (frOpn3+DT) (revised **Fig. 7e** and **Result section line: 226**).

In addition, the actual tumor volume of all experimental groups (frOpn1/frOpn3±DT) is also now demonstrated for the same time points (revised Fig. 7e and Result section lines: 226-7). The revised Fig. 7e presents the kinetics of tumor growth up to 10 days after the 1st frOpn1 injection, instead of 6 days shown in the original submitted manuscript. As can be observed in the revised Fig. 7e, pDC depletion does not affect tumor growth independently of frOpn1 (and Result section lines: 229-30). Thus, the data added suggest that the tumor suppressing effect of frOpn1 is indeed pDC-mediated.

Revised Figure 7e:

According to the reviewer’s recommendations, we felt that the pDC-depleted experimental groups (frOpn3+DT and frOpn1+DT) should be added also in Figure 7f. Therefore, the revised Figure 7f showed that there was no enhancement on IFN- γ -expressing intratumoral CD8⁺ T cell numbers in pDC-depleted groups. In addition, the measurements after administration of both frOpn1 and frOpn3 were similar in pDC-depleted groups, pointing that this increase after frOpn1 treatment was indeed pDC-mediated. By comparing between ±DT-treated groups it is now revealed that pDC depletion did not affect IFN- γ -expressing intratumoral CD8⁺ T cell numbers independently of frOpn1 (revised Fig. 7f and Result section lines 233-9).

Revised Figure 7f:

To corroborate our findings, we utilized control PBS-treated and DT-treated groups in the revised **Figures 7b-d** (and **Result section lines 211-3, 217**). Addition of the DT control group up to 6 days after the 1st frOpn1 injection, demonstrated also the efficacy of intratumoral pDC depletion in DT-treated mice (revised **Fig. 7b, d** and **Result section lines 220-2**).

Revised Figure 7b:

Revised Figure 7c:

Revised Figure 7d:

10. The authors propose a mechanistic model by which *Opn/Itga4* activates PI3K/Akt/mTOR signaling, which in turn results in increased IRF3 activation and subsequent IFN β production. However, while their graphical abstract suggests a direct post-translational activating effect of mTOR on IRF3, their data in fact suggest that PI3K/mTOR signaling somehow promotes *Irf3* transcription in response to *Opn* (Fig. 5F, and also Fig. 3A). This seems to be consistent with the apparent general increase in IRF3 protein level observed in their confocal experiments. To clarify this, I would urge the authors to carefully evaluate the levels of IRF3 mRNA, phospho-IRF3, and total IRF3 over a detailed time-course of several hours to better establish whether IRF3 is actually activated by PI3K/mTOR at a post-translational level, or if increased IRF3 activity is driven by upregulated transcription.

We are grateful to the reviewer's suggestions for the best representation of our findings and enlightenment of the proposed mechanism. As correctly captured by the reviewer, our manuscript proposes a mechanistic model where *Opn/Itga4* activates PI3K/Akt/mTOR signaling, which results in increased IRF3 activation and subsequent IFN- β production. We apologize for not explaining in detail the meaning of the dotted-line connecting mTOR to IRF3 in the legend of our original graphical abstract. Our purpose was not to suggest a direct post-translational activation effect of mTOR on IRF3, thus a dotted line was used to connect PI3K/Akt/mTOR-phosphorylated P70S6K and cytoplasmic IRF3, as further experiments are still necessary to certify this pathway. The meaning of the dotted line it is clarified now in the revised legend of the graphical abstract (**new Fig. 8b**).

We truly appreciate the reviewer's suggestion of evaluating the levels of IRF3 mRNA, phospho-IRF3, and total IRF3 over a detailed time-course to answer whether IRF3 is activated directly at a post-translational level, or if increased IRF3 activity is driven by upregulated transcription in response to *Opn*. Thus, we utilized immunofluorescence staining to monitor IRF3 translation and nuclear translocation upon *Opn* stimulation of *Irfar1*^{-/-} pDCs. The new graphs show the confocal analysis over a 6 hrs period with measurements taken every half hour after fr*Opn* treatment (**new Fig. 8a**). Cytoplasmic IRF3 expression starts rising from the baseline at 1.5 hrs, peaking at 2.5 hrs and dropping at 3hrs, which coincides with the 3hr peak of nuclear translocation (**new Fig. 8a**). Nuclear translocation of IRF3 starts rising at 2.5 hrs, peaking at 3 hrs (timepoint demonstrated also in **revised figures 3b-c, 4c, 5g**) and at 3.5hrs almost returns to baseline expression to remain in these levels (**new Fig. 8a**). Similarly, *Irf3* mRNA transcription starts rising at 1.5 hrs of treatment and reaches the highest levels at 3hrs (demonstrated also in figures 3a, 5f), afterwards it decreases and remains unchanged for 9 hrs (**new Fig.8a** and data not shown).

The time-course observations demonstrated that *Opn*-mediated enhanced *Irf3* transcription from 1.5 to 3.5 hrs, resulting in enhanced cytoplasmic IRF3 expression (peak at 2.5 hrs) followed by IRF3 phosphorylation/nuclear translocation (peak at 3hrs), (**new Fig. 8a**). The significant boost in *Irf3* transcription (2.5-3 hrs) occurred concomitantly with the enhancement of cytoplasmic IRF3 production, indicating that newly translated IRF3 protein was translocated to the nucleus and not the pre-existing cytoplasmic IRF3 (**new Fig. 8a**). Supportively, the timeframe for mTOR activation in our setting starts approximately as early as 1-1.5hrs, as both Akt and Tsc2 phosphorylation took place within 30 min (**revised Fig. 5a**) and p70S6K phosphorylation started to exhibit statistically significant differences within 1hr after fr*Opn* addition (**revised Fig. 5d**). Therefore, we understand that IRF3 phosphorylation is not a result of a direct activating effect of mTOR, as the nuclear translocation of IRF3 takes place 2.5 hrs after fr*Opn* addition (**new Fig. 8a**). Whereas, for a direct post-translational activating effect of mTOR we would expect a swift response at a time point earlier than 2.5hrs. The above data suggest that the activation of PI3K/Akt/mTOR increases baseline values of IRF3 by upregulated transcription and it is not activated by mTOR directly at a post-translational level. We demonstrated our findings in the **Result section lines: 180-195** and **Discussion section lines: 364-76**.

Fig. 8a:

Fig. 8b:

11. Mechanistically, the authors do not show through direct loss-of-function experiments that IRF3 is actually necessary for the induction of IFN β in response to Opn. Given that other pathways can induce low-level IFN β production in the absence of strong IRF3/7 signaling (such as NF κ B), the authors should consider approaches to establish whether IRF3 is in fact the critical driver of IFN β (e.g. using pDC from IRF3-deficient mice), or if it is another pathway downstream of PI3K signaling.

We feel that the reviewers' suggestion for direct loss-of-function experimentation would significantly assist in the elucidation of our proposed mechanism. Initially we thought to utilize the IRF3-inducible KO mice, but unfortunately, we were informed that the procedures required for their acquisition would take approximately 4-6 months (order, transportation, restrictions etc.), therefore the timeline of our experiments would exceed the timeframe of JCB revisions. This led us to try a timewise solution of a silencer siRNA-mediated deletion of IRF3 in pDCs. The efficiency of IRF3 deletion with this method (si-IRF3, Santa Cruz Biotechnology) was previously tested (Al Moussawi *et al.* *PLoS Pathog* 2010). For siRNA delivery into pDCs, we utilized a lipid-based reagent

(DOTAP) that was proven to prevent possible induction of pDC death or activation (Smith *et al. Scientific Reports* 2016). We successfully knocked-down 86% of IRF3 mRNA in pDCs, as assessed by RT-PCR (**new Fig. S1b**). Efficient IRF3 protein reduction was confirmed by western blot (**new Fig. S1c**). We also verified that IRF-3 silencing in pDCs significantly suppressed IFN- β production upon poly (I:C) stimulation, providing a validation of loss-of-function (**new Fig. S1d**).

After frOpn1 treatment, we measured diminished ability of IRF3-silenced pDCs to enhance both mRNA expression and secretion of IFN- β , indicating that IRF3 directly mediates the Opn-induced IFN- β production observed via integrin α 4 (**new Fig. 3g**). We mention these findings in **Result section lines: 134-40**, **Method section lines: 515-9** and **Discussion section lines: 358-63**.

Figure 3g:

Minor concerns:

1. Fig. 3A: what are expression values normalized to ("relative" to what)?

In the original version of our manuscript, we mentioned each real-time PCR performed was normalized to *Hprt* expression (previous Method Section lines: 402). In the revised **Fig. 3** and in all the remaining figures containing RT-PCR data, we now add this information (revised **Figure Legends 1-5**).

2. Alpha 4 integrin (and beta 1 integrin) is expressed at high levels on other innate immune populations including monocytes, macrophages and conventional DC. Does Opn uniquely induce IFN β production in pDC, or does this mechanism also exist in other myeloid populations?

As mentioned by the reviewer, *Itga4* is also expressed in other innate cell populations. We and others have studied the role of Opn/*Itga4* axis in cell populations such as monocytes, macrophages, neutrophils and cDCs in various experimental settings (Lund *et al. JCBiochem* 2013, Uede *et al. Pathol Int* 2011, Kourepini *et al. PNAS*

2014). In these myeloid populations, the findings indicate that the role of Opn/Itga4 axis is mainly on cell chemotaxis, recruitment and survival (unpublished data from our lab and Lund *et al. J C Biochem* 2013, Uede *et al. Pathol Int* 2011, Kourepini *et al. PNAS* 2014). However, the induction of *Ifnb* expression by $\alpha 4$ integrin activation is a mechanism uniquely found in pDCs. We add now this information in the **Discussion section lines: 292-8.**

3. The IFNAR1 gene name is frequently misspelled as "IFNRA1" and should be corrected

We thank the reviewer for noticing this typographic error that is now corrected.

Response to Reviewer #2:

(Reviewer's comments are in **bold** characters.)

Comments:

In this manuscript, the authors describe a novel axis that promotes a low level of IFN- β production by plasmacytoid dendritic cells in the absence of inflammatory signals. This axis is mediated by PI3K/mTOR signaling upon binding of the SLAYGLR motif of osteopontin to $\alpha 4$ integrin. This is a potentially interesting finding with possible implications for pDC-mediated immune responses (especially anti-tumor), but there are several issues that must be addressed.

We sincerely thank the reviewer for thoroughly reviewing our manuscript as well as for the positive comments and valuable suggestions, which were incorporated to our revised manuscript. Of note, most comments and suggestions were consistent among reviewers. Below is a point-by-point reply to the reviewer's comments.

Major:

1. The peptides used by the authors for engagement of $\alpha 4$ integrin are very short and it is not clear how strongly this binding resembles a ligation by actual products of thrombin-mediated osteopontin cleavage. Also, Figure 5a and d lack multiple controls used in other experiments. Therefore, it is unclear to what extent the agonistic frOpn1 peptide triggers relevant signaling.

The reviewer raised a key question for the comprehension of the function of the particular Opn fragments used in our experiments. In our previous studies, we had extensively analyzed the capacity of specific fragments to effectively ligate $\alpha 4/\alpha 9$ integrins, where frOpn₁₃₄₋₁₅₃ (frOpn1) fully resembles ligation by thrombin cleaved Opn (Kourepini *et al. PNAS* 2014, Alissafi *et al., JI* 2018), but it is plausible that this information should be reported again in the current manuscript. We apologize for not analyzing this important aspect in our previous discussion. Thus, we now added the relevant information in the revised version (**Discussion Section lines: 311-29**). The Opn peptide used for $\alpha 4$ integrin engagement (frOpn1 fragment: scrambled in RGD, intact in SLAYGLR) represents the thrombin cleaved fragment of Opn with exposed cryptic SLAYGLR domain (Yokosaki *et al., J Biol Chem* 1999). Thrombin cleaves Opn after the SLAYGLR amino-acid (aa) sequence in two fragments, one containing the amino (N)-terminal and the other the carboxy (C)-terminal part. By using a 20aa of the N-terminal part (icosamer), we restricted the reactive domains and avoided aa sequences that may contain sites capable of interacting with other molecules and possibly masking the effects of specific integrin binding. For example, next to the N-terminal there is a calcium binding domain that could alter the effect of SLAYGLR domain of interest, as calcium have been found to suppress cell adhesion to Opn by attenuating binding affinity to integrins (Hu *et al. JCBiochem* 1995). In addition, located in the N-terminal part of Opn and next to the calcium site, there is a CD44 binding domain (Buback *et al., Exp Derm* 2009). Beside the N-terminal product, thrombin cleavage of Opn produces a fragment containing the C-terminal, which is also capable of interacting with several CD44 variants (Weber *et*

al., *Science* 1996; Wang et al., *Cytokine & growth factor reviews* 2008) and possibly affecting pDC function. We used the shortest active peptide version in order to dissect the role of SLAYGLR domain which binds only to $\alpha 4$ and $\alpha 9$ integrins. In our previous studies, as well as in other settings, this icosamer demonstrated to have greater efficiency compared to full length or to thrombin cleaved Opn (Kourepini et al., *PNAS* 2014; Albertsson et al., *JNeuroinfl* 2014; Doyle et al., *JCBF Metabol* 2008). It is thus possible that interaction of Opn with other molecules (e.g CD44), interferes with certain Opn effects such as pDC function. Of note, the use of icosamer is usually better tolerated upon therapeutic use in several disease models tested when compared to larger fragments, or to the one that contains the complete N-terminal domain (Zhou et al., *Journal of Cellular and Mol Med* 2020; Albertsson et al., *JNeuroinfl* 2014; Doyle et al., *JCBF Metabol* 2008; Jin et al., *Mol Neurobiol* 2016).

Therefore, as the reviewer correctly noticed, without the demonstration of control groups in Figures 5a and d, it was not so clear to what extent the agonistic frOpn1 peptide triggers relevant signaling. In our original version we had only mentioned the use of control groups for Figures 5a and d in the method section (lines 410-1). As also suggested by the other reviewer (6th comment), in order to clarify the capacity of frOpn1 to drive the changes in pAkt and p70S6K, we included timepoint matched samples (PBS-treated pDCs) as the most appropriate control groups in both **Figures 5a** and **5d**.

Utilization of the control pDCs pointed out that phosphorylation of Akt was indeed frOpn1-driven (revised **Fig. 5a**). It was observed that p70S6 kinase was phosphorylated through culture time, however frOpn1 treatment enhanced significantly the levels of phosphorylation for both Ser371 and Thr389 when compared to the respective time-points of the control group (revised **Fig. 5d**). We also added matched quantitative graphical plots for each protein demonstrated in western blots, normalized to β -actin (revised **Fig. 5a, 5d**). We now add these results also in **Result section lines: 158-9, 170-3**.

In order to corroborate these findings, we measured phosphorylated Tsc2 protein in PBS- and frOpn-treated pDC groups. We added a western blot image and a quantification bar graph, demonstrating significantly raised Tsc2 protein phosphorylation after frOpn1-treatment of pDCs, with highest levels at 30 min which coincides with the time point where Akt is also highly phosphorylated (revised **Fig. 5a**). We now add these data in **Result section lines: 161-3, Method section line: 504 and Discussion section lines: 335-6, 346-9**.

Revised Figure 5a:

Revised Figure 5d:

2. In Figure 3c, the authors quantify the nuclear/cytoplasmic ratio of IRF3 MFI. This should be done for all graphs depicting IRF3 MFI in order to better account for likely background expression in different cells/experiments.

We are grateful for this comment which is related to the suggestions of the other reviewer. The reviewer fairly commented that the adopted format for demonstrating IRF3 quantity by a dot plot depicting nuclear/cytoplasmic MFI in Figure 3c should be used in all confocal figures. As suggested by the reviewer, we calculated the ratio of nuclear/cytoplasmic values in all figures containing IRF3 MFI, but this proved not to be the best representation, as in many samples cytoplasmic values were diminished at 3 hrs of measurement. There were several cells which showed the highest percentage of IRF3 located in the nucleus compared to a very small MFI value in the cytoplasm, thus resulting in a calculated ratio with plasmatic high numbers. Therefore, as suggested for comparable data among experiments and the most accurate result representation, the confocal analysis graph showing the ratio nuclear/cytoplasmic IRF3 MFI was replaced by a detailed dot plot demonstrating total IRF3 (cytoplasmic + nuclear) signal intensity in individual cells, compared to nuclear IRF3 (revised **Fig. 3c, Result section lines: 123-5**). We feel that with this alternative representation, IRF3 MFI and likely background expression of different cells/experiments can be comparable. For that reason, we also replaced in Figures 3d, 4c and 5g the bar graphs showing MFI of nuclear IRF3, with dot plots demonstrating nuclear and total MFI in individual cells (revised **Figures 3c, 3d, 4c and 5g**). This comment helped us to find and correct a typographical mistake in the axis numbers of the previous version of **Fig. 5g**.

Plot in revised Figure 3c:

Plot in revised Figure 3d:

Plot in revised Figure 4c:

Plot in revised Figure 5g:

3. In Figure 6, the authors claim that the enhancement of IFN- α production by pre-treatment with frOpn1 following CpG-A stimulation depends on IFNAR signaling triggered by IFN- β . However, the actual dependence of this effect on IFN- β is not directly addressed. Instead, only a complete dependence of any IFN- α production on the presence of IFN- α/β receptor is shown. The authors should block IFN- β in order to conclude that the "exaggerated responses to TLR triggering" depend on Opn/SLAYGLR-mediated IFN- β production.

We appreciate the reviewer's concerns on the important aspect on whether IFN- α production by pre-treatment with frOpn1 following CpG-A stimulation depends actually on IFN- β . As proposed by the reviewer, in order to address whether the enhancement of IFN- α production following CpG-A treatment depend on frOpn1-mediated IFN- β production, we added a pDC group where we neutralized IFN- β by a blocking antibody (revised **Fig. 6, Result section lines: 202-5**). Our results revealed that indeed the enhancement of IFN- α production by pre-treatment with frOpn1 following CpG-A stimulation depends on IFNAR signaling triggered by IFN- β . We also added this information that further elucidates our findings in the **Discussion section lines 280-2**.

Revised Figure 6:

4. Figure 7 lacks basic controls such as PBS. Also, only one group in Figure 7e has a DTR control. Moreover, such DTR controls are missing from crucial experiments showing effects on T cells in Figure 7f. In this Figure 7f, the authors show an increased number of intratumoral IFN- γ producing CD8+ T cells following frOpn1 treatment in comparison to frOpn3. However, this effect is not necessarily dependent on pDCs and could be possibly mediated by direct effects on T cells. Therefore, depletion of pDCs with DT (as in Figure 7e) is necessary for this conclusion. Further, the authors only show relative tumor volume 5 days after frOpn treatment (14-15 days total), which is a short time frame for the B16-F10 model. The authors should include the actual total tumor volume and follow the tumors to a later time point (such as day 25).

Reviewer's observation is to the point; therefore, we now added the missing controls in the relative graphs of the revised **Figure 7**. For better understanding of our findings, we utilized control groups (PBS-treated and DT-treated mice) which are now depicted in revised **Figures 7b-d** and **Result section lines 211-3, 217**. Addition of the DT control group up to 6 days after the 1st frOpn1 injection, demonstrated also the efficacy of intratumoral pDC depletion in DT-treated mice (revised **Fig. 7b, d** and **Result section lines 220-2**).

Revised Figure 7b:

Revised Figure 7c:

Revised Figure 7d:

We are grateful for the crucial comments supporting the understanding of our findings on the development of melanoma, thus the impact of pDC depletion in the absence of frOpn1 treatment is now also analyzed and it is presented in the revised manuscript. This suggestion is in agreement among the two reviewers (9th comment of the other reviewer). In **Figure 7e** where the relative tumor volume is demonstrated, we added a group of pDC-depleted mice treated with the inactive peptide (frOpn3+DT) (revised **Fig. 7e** and **Result section lines: 226**). In addition, the actual tumor volume of all experimental groups (frOpn1/frOpn3±DT) is also now demonstrated for the same time points (revised **Fig. 7e** and **Result section lines: 226-7**). Interestingly, pDC depletion did not affect tumor growth independently of frOpn1 (revised **Fig. 7e** and **Result section lines: 229-30**). Thus, the new data suggest that the tumor suppressing effect of frOpn1 is indeed pDC-mediated.

The reviewer suggests that the kinetics of tumor growth should be presented at later time point. The timeframe presented previously in **Fig. 7e** demonstrate results until 14-15 days after B16.F10 inoculation (day 6 of frOpn1 treatment), which is a typical timeframe described in other studies for this specific melanoma model (Drobits et al. *JCI* 2012). However, as proposed by the reviewer, the revised **Fig. 7e** now demonstrates the results up to day 19 (after the B16.F10 inoculation), which corresponds to day 10 of frOpn1 treatment. For more consistent data representation, similar timepoints are also presented in the new graph showing the actual tumor volume (revised **Fig. 7e**). We apologize for not extending the graph presentation up to day 25 as suggested, but this option was not feasible in our experimental setting. Beyond day 19, the mice in the control groups were prone

to high mortality with large tumors and required to be euthanized, according to the Ethics Protocol approved by the Institutional Committee of Protocol Evaluation in association with the Hellenic Directorate of Agriculture and Veterinary Policy and in agreement to the guidelines for this particular melanoma model (mentioned in method lines 351-4, now in 434-7, Overwijk *et al. Curr Prot Immunol* 2001). Briefly, subcutaneously injected B16.F10 cells form a palpable tumor in 5 to 10 days and grow to a 1cm³ tumor in 14-21 days. When allowed to grow larger, the tumors often become necrotic in the center and begin to ulcerate or bleed, thus, it is advisable to sacrifice the mice before this point (Overwijk *et al. Curr Prot Immunol* 2001).

Revised Figure 7e:

As recommended by the reviewer, pDC-depleted experimental groups (frOpn3+DT and frOpn1+DT) were added in **Figure 7f**. The revised **Figure 7f** clarified that IFN- γ -expressing intratumoral CD8⁺ T cell numbers were not enhanced in pDC-depleted groups and their levels were similar in frOpn1+DT and frOpn3+DT groups, implying that this increase after frOpn1 treatment was pDC-mediated. Conclusively, these data reveal that the enhancement of IFN- γ -expressing intratumoral CD8⁺ T cell numbers is fully dependent on pDCs and it is not mediated by direct effects on T cells. By comparing \pm DT-treated groups we were also able to demonstrate that pDC depletion did not affect IFN- γ -expressing intratumoral CD8⁺ T cell numbers independently of frOpn1 (revised **Fig. 7f** and **Result section lines 233-9**).

Revised Figure 7f:

Minor:

1. Throughout the manuscript, the authors refer to the type-I interferon- α/β receptor gene as *Ifnra1*, but the proper gene name is *Ifnar1*. This should be corrected.

Line 168: Figure 8 is mistakenly called out.

We thank the reviewer for noticing these typographical errors that are now corrected.

2. Figure 3b: the representative immunoblot is not clear, which makes it difficult to interpret.

We appreciate the reviewer's concern on the representation of this result. For the western blot in Figure 3b, an ultra-high sensitivity developing reagent was used (previously mentioned in method section line 409). Thus, in order to limit background appearance, we now provide a clearer/higher definition picture as requested by the reviewer (**Fig. 3b**). To corroborate these data, we have also now added bar graphs showing western blot analysis for protein levels of total IRF-3 and p-IRF3, normalized to β -actin, which are both significantly increased upon frOpn1 addition when compared to the controls (**Fig. 3b, Result section lines: 121-3**).

Revised Figure 3b:

3. Figure 3d-e: Blockade of α 9 integrin appears to increase IRF3 MFI, but it not shown to be statistically significant. Further, blockade of α 9 integrin increases *Ifnb* expression. The authors discuss these results in lines 130-133 and only say that α 9 integrin does not decrease IRF3 nuclear translocation or *Ifnb* expression. Can the authors comment on this?

As the reviewer correctly suggested, we should discuss further the increases of IRF3 MFI and *Ifnb* expression observed after blockade of α 9 integrin, that both exhibited statistical significance. To support the discussion of our findings, we replaced the bar graphs in **Fig. 3d** with a dot plot graph of single cell values. The significance in the increase of IRF3 MFI and *Ifnb* expression in pDCs after α 9 integrin blockade is now clearly represented (revised **Fig. 3d**). These results led to the question whether α 9 integrin blockade alone induces *Ifnb* expression, which is consistent with the other reviewer's 2nd comment. In order to shed light on this, the revised manuscript now includes the corresponding control groups of pDCs treated with either α 9 integrin Ab or Ig Ab without frOpn1 addition (PBS controls, gray bars, revised **Fig. 3e**, right graph). By demonstrating these controls, we clarified that blockade of α 9 integrin alone does not cause a significant increase in *Ifnb* expression in pDCs, but it is the presence of frOpn1 which is required for this increment. In support to this phenomenon observed, upon α 9 integrin blockade we measured a 3-fold enhancement in *Itga4* mRNA expression in pDCs (data not shown). Considering that α 4 and α 9 integrins are the known receptors for frOpn1, our data suggest that α 9 integrin blockade results in binding of frOpn1 to available and elevated α 4 integrin, which further boosts *Ifnb* induction. We now discuss this point in the **Discussion section lines: 300-5**.

We now add also the corresponding control groups for either α 4 integrin Ab or Ig Ab, without frOpn1 (PBS controls, gray bars, revised **Fig.3e** left graph).

Plot in revised Figure 3d:

Revised Figure 3e:

4. Figure 5c: It is unclear what the statistics are comparing.

Thank you for this observation. We now provide analytic statistics for Figure 5c.

Revised Figure 5c:

Note for both Reviewers: We have also added new statistical analyses as we performed two-way ANOVA instead of Student's t-test when comparing more than 2 samples and we corrected for the multiplicity of tests with Bonferroni correction. We mention the strategy in the **Materials and Methods Section lines 520-24.**

September 22, 2021

RE: JCB Manuscript #202102055R

Dr. Evangelia Kourepini
Biomedical Research Foundation of the Academy of Athens
Center for Basic Research
4 Soranou Efessiou Street
Athens 11527
Greece

Dear Dr. Kourepini,

Thank you for submitting your revised manuscript entitled "An integrin axis induces IFN- β production in plasmacytoid dendritic cells." We would be happy to publish your paper in JCB pending final revisions necessary to meet our formatting guidelines and to address two remaining minor points from Reviewer #2 (see details below).

A. MANUSCRIPT ORGANIZATION AND FORMATTING:

Full guidelines are available on our Instructions for Authors page, <https://jcb.rupress.org/submission-guidelines#revised>. **Submission of a paper that does not conform to JCB guidelines will delay the acceptance of your manuscript.**

- 1) Text limits: Character count for Articles is < 40,000, not including spaces. Count includes title page, abstract, introduction, results, discussion, and acknowledgments. Count does not include materials and methods, figure legends, references, tables, or supplemental legends.
- 2) Figures limits: Articles may have up to 10 main text figures.
- 3) Figure formatting: Scale bars must be present on all microscopy images, including inset magnifications. Molecular weight or nucleic acid size markers must be included on all gel electrophoresis. Please add markers to Figures 3B, 5A&D, S1C, and scale bars to Figures 3C, 4C, 5G, S1A.
- 4) Statistical analysis: Error bars on graphic representations of numerical data must be clearly described in the figure legend. The number of independent data points (n) represented in a graph must be indicated in the legend. Statistical methods should be explained in full in the materials and methods. For figures presenting pooled data the statistical measure should be defined in the figure legends. Please also be sure to indicate the statistical tests used in each of your experiments (both in the figure legend itself and in a separate methods section) as well as the parameters of the test (for example, if you ran a t-test, please indicate if it was one- or two-sided, etc.). Also, if you used parametric tests, please indicate if the data distribution was tested for normality (and if so, how). If not, you must state something to the effect that "Data distribution was assumed to be normal but

this was not formally tested."

5) Materials and methods: Should be comprehensive and not simply reference a previous publication for details on how an experiment was performed. Please provide full descriptions (at least in brief) in the text for readers who may not have access to referenced manuscripts. The text should not refer to methods "...as previously described."

6) For all cell lines, vectors, constructs/cDNAs, etc. - all genetic material: please include database / vendor ID (e.g., Addgene, ATCC, etc.) or if unavailable, please briefly describe their basic genetic features, even if described in other published work or gifted to you by other investigators (and provide references where appropriate). Please be sure to provide the sequences for all of your oligos: primers, si/shRNA, RNAi, gRNAs, etc. in the materials and methods. You must also indicate in the methods the source, species, and catalog numbers/vendor identifiers (where appropriate) for all of your antibodies, including secondary.

7) Microscope image acquisition: The following information must be provided about the acquisition and processing of images:

a. Make and model of microscope

b. Type, magnification, and numerical aperture of the objective lenses

c. Temperature

d. Imaging medium

e. Fluorochromes

f. Camera make and model

g. Acquisition software

h. Any software used for image processing subsequent to data acquisition. Please include details and types of operations involved (e.g., type of deconvolution, 3D reconstitutions, surface or volume rendering, gamma adjustments, etc.).

8) References: There is no limit to the number of references cited in a manuscript. References should be cited parenthetically in the text by author and year of publication. Abbreviate the names of journals according to PubMed.

[[Check and make sure that the authors did not include a reference list in the supplementary methods - we do not allow supp references. If so, you should add a line here asking them to remove it and add any non-duplicate references to the main reference list.]]

9) Supplemental materials: There are strict limits on the allowable amount of supplemental data. Articles/Tools may have up to 5 supplemental figures and 10 videos. Please also note that tables, like figures, should be provided as individual, editable files. A summary of all supplemental material should appear at the end of the Materials and methods section. Please include one brief sentence per item.

10) eTOC summary: A ~40-50 word summary that describes the context and significance of the findings for a general readership should be included on the title page. The statement should be written in the present tense and refer to the work in the third person. It should begin with "First author name(s) et al..." to match our preferred style.

11) Conflict of interest statement: JCB requires inclusion of a statement in the acknowledgements regarding competing financial interests. If no competing financial interests exist, please include the following statement: "The authors declare no competing financial interests." If competing interests are declared, please follow your statement of these competing interests with the following

statement: "The authors declare no further competing financial interests."

12) A separate author contribution section is required following the Acknowledgments in all research manuscripts. All authors should be mentioned and designated by their first and middle initials and full surnames. We encourage use of the CRediT nomenclature (<https://casrai.org/credit/>).

13) ORCID IDs: ORCID IDs are unique identifiers allowing researchers to create a record of their various scholarly contributions in a single place. At resubmission of your final files, please consider providing an ORCID ID for as many contributing authors as possible.

B. FINAL FILES:

Thank you for this interesting contribution, we look forward to publishing your paper in Journal of Cell Biology.

Sincerely,

Ira Mellman, Ph.D.
Monitoring Editor

The Journal of Cell Biology

Dan Simon, Ph.D.
Scientific Editor
The Journal of Cell Biology

Reviewer #1 (Comments to the Authors (Required)):

The authors have addressed all of my comments/concerns. No further revisions required.

Reviewer #2 (Comments to the Authors (Required)):

The authors fully responded to the major questions raised by this reviewer. Two minor points remaining:

- 1) the correct spelling of *lfnar1* must be used everywhere, the authors missed it in multiple places (Figure 1B and C, and in Figure 8B).
- 2) the designation of "blocking antibody" should clearly used i.e. anti-XY.